# Sustained increase in suspended sediments near global river deltas over the past two decades

Xuejiao Hou[1,2], Danghan Xie [2,3], Lian Feng [4] ✉, Fang Shen[5] &
Jaap H. Nienhuis [2] ✉

River sediments play a critical role in sustaining deltaic wetlands. Therefore, concerns are raised about wetlands' fate due to the decline of river sediment supply to many deltas. However, the dynamics and drivers of suspended sediment near deltaic coasts are not comprehensively assessed, and its response to river sediment supply changes remains unclear. Here we examine patterns of coastal suspended sediment concentration (SSC) and river sediment plume area (RPA) for 349 deltas worldwide using satellite images from 2000 to 2020. We find a global increase in SSC and RPA, averaging +0.46% and +0.48% yr$^{-1}$, respectively, with over 59.0% of deltas exhibiting an increase in both SSC and RPA. SSC and RPA increases are prevalent across all continents, except for Asia. The relationship between river sediment supply and coastal SSCs varies between deltas, with as much as 45.2% of the deltas showing opposing trends between river sediments and coastal SSCs. This is likely because of the impacts of tides, waves, salinity, and delta morphology. Our observed increase in SSCs near river delta paints a rare promising picture for wetland resilience against sea-level rise, yet whether this increase will persist remains uncertain.

Suspended sediments in the coastal ocean play a crucial role in maintaining coastal wetlands, marine ecosystems[1–4], and deltas[5]. In recent decades, various global-scale assessments have highlighted a decline in sediment supply from many rivers due to the construction of river dams[6–10]. Such a decline is often assumed to limit coastal suspended sediment concentration (SSC) and subsequently leads to increased risks of coastal wetland loss[2,11,12]. However, evidence of a link between river sediment supply reduction and coastal SSC decline is mixed. For example, an extensive compilation of coastal suspended sediment accumulation rates shows that coastal sediment deposition increased in the 20th century despite the construction of river dams[13]. A possible explanation is that coastal SSCs are also affected by feedbacks between coastal hydrodynamics (e.g., river flow, tides, and waves) and sediment transport (e.g., suspension, erosion, deposition, and movement)[14–17]. As such, a better understanding of the long-term variations of coastal SSCs and their controls is essential for accurately assessing the future of coastal communities.

Previous evaluations of global coastal SSC changes have been conducted using satellite images, revealing a decrease in SSCs in many coastal areas[18,19]. However, these findings were based on a limited monitoring period (e.g., from 2003 to 2012), and the underlying factors driving these changes remain unclear, impeding our comprehensive understanding of global deltaic SSC changes over recent decades. Nevertheless, for individual deltas such as the Yangtze and Mekong, numerous studies have thoroughly investigated the long-term trends in SSCs and their potential influences, revealing a decrease

[1]School of Geospatial Engineering and Science, Sun Yat-Sen University, Guangzhou, China. [2]Department of Physical Geography, Utrecht University, Utrecht, the Netherlands. [3]Department of Earth and Environment, Boston University, Boston, MA, USA. [4]School of Environmental Science and Engineering, Southern University of Science and Technology, Shenzhen, China. [5]State Key Laboratory of Estuarine and Coastal Research, East China Normal University, Shanghai, China. ✉e-mail: fengl@sustech.edu.cn; j.h.nienhuis@uu.nl

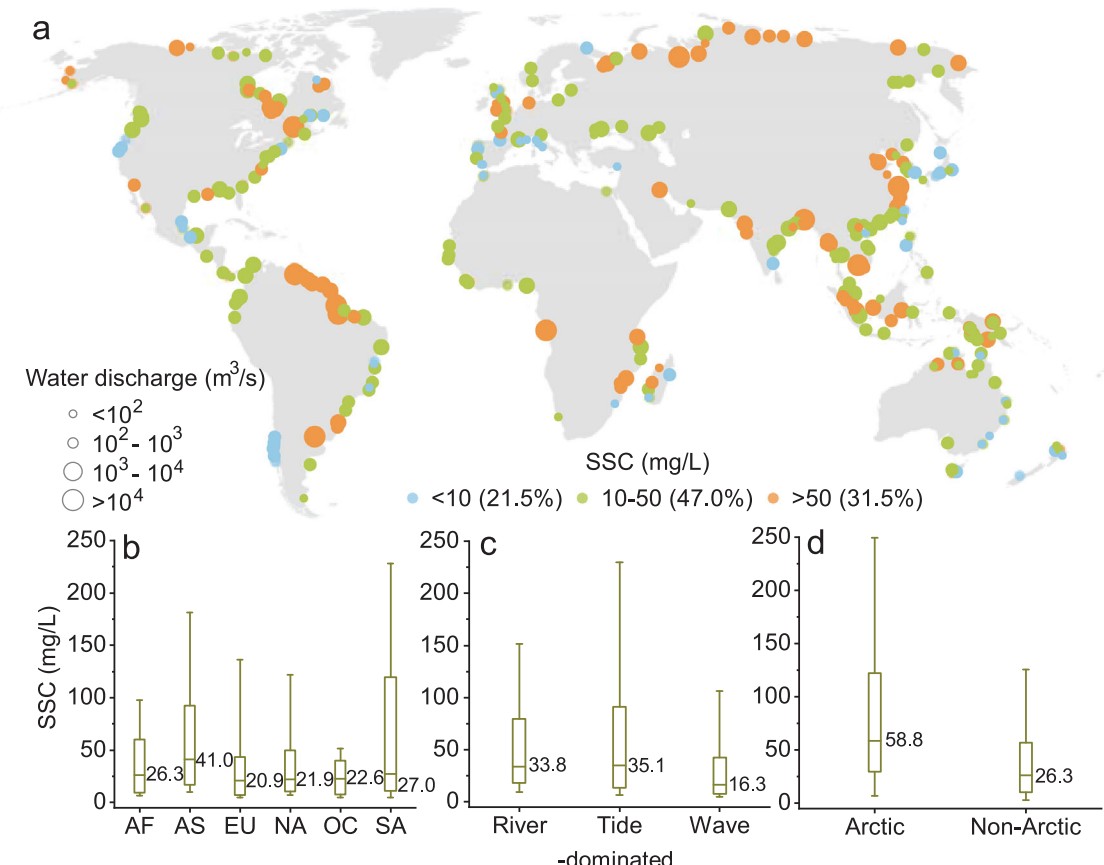

**Fig. 1 | Global pattern of coastal suspended sediment concentration (SSC) between 2000 and 2020. a** The long-term mean SSC (unit: mg/L) for 349 deltas. The different circle sizes represent varying magnitudes of water discharge, while the different circle colors indicate different levels of SSC. **b–d** Box plots of long-term mean SSC in different continents (AF:Africa, AS:Asia, EU:Europe, NA:North America, OC:Oceania, SA:South America) (**b**), various delta morphologies (**c**), and the Arctic (with latitude >50°N) and non-Arctic deltas (**d**). The box plots in **b–d** show the distributions (10, 25, 50, 75, and 90% values) of SSC.

in SSCs attributed to dam construction and subsequent decline in river sediment supply[20,21]. However, it remains uncertain whether these declines hold globally, and what the relevant drivers might be.

To address these knowledge gaps, we utilized globally available satellite data between 2000 and 2020 to obtain a monthly record of coastal SSC near 349 major river deltas. We attempt to answer three fundamental questions: (i) What are the spatial and temporal patterns of coastal SSC near river deltas worldwide over the past two decades? (ii) How does SSC respond to the changes in river sediment supply? (iii) What are the possible other controls on coastal SSC near river deltas?

## Results

### Mapping coastal SSC near global river deltas

We mapped coastal surface SSC near deltas using 500-m resolution moderate-resolution imaging spectroradiometer (MODIS) Terra and Aqua 8-day surface reflectance (SR) products. This mapping employed a precise coastal SSC retrieval algorithm (root mean square error = 24.9%), which we adapted from a global algorithm developed by ref. 22 based on our collected coastal in situ SSC measurements (Supplementary Fig. 1) (see Methods). The satellite-derived SSCs generated by this algorithm are consistent with in situ measured SSC, SSC from OLCI products (with a spatial resolution of 4 km), and SSC documented in other local studies (Supplementary Fig. 1 and Supplementary Tables 1, 2). Using this algorithm and 0.58 million MODIS 8-day SR image composites from 2000 to 2020, we obtained long-term records of coastal surface SSCs around 349 major river deltas covering six continents. These examined rivers account for 64.2% of the global total water discharge[5], covering a wide range of delta

morphologies (Supplementary Fig. 2), including river-dominated ($n = 136$), tide-dominated ($n = 87$), and wave-dominated ($n = 126$) (from ref. 5). In addition to the SSC, we investigated the river sediment plume area (RPA, unit: km$^2$) as the extent where the monthly mean SSC over an SSC threshold determined for each river mouth (see Methods) (Supplementary Fig. 3). This metric was then used to capture the spatial variations of coastal sediment plumes at the sea surface. Our observations primarily reflect surface SSC and RPA, SSC dynamics deeper down in the water column may be different and were not considered here.

We find that the long-term mean surface SSCs displayed substantial spatial heterogeneity among the 349 deltas (Fig. 1a), varying from 2.8 (Tamar, Australia) to 379.7 mg/L (Mahi, India), with a median of 29.1 mg/L. Among all the deltas examined, 21.5% exhibited mean SSCs below 10 mg/L, while 31.5% displayed mean SSCs over 50 mg/L. Deltas with higher SSCs, particularly those with water discharge exceeding 10,000 m$^3$/s, were predominantly located in Asia (median of 41.0 mg/L) (Fig. 1b). This higher SSC can be attributed to the substantial sediment loads from large rivers such as the Yangtze, Mekong, Ganges, and certain Arctic rivers (e.g., Kolyma and Lena) in this region (Fig. 1d). On the other hand, deltas with lower SSCs were mainly found in Europe, with a median SSC of 20.9 mg/L. Additionally, the long-term mean SSC also varied among different delta morphologies (Fig. 1c), with the highest SSC observed in tide-dominated deltas (median of 35.1 mg/L), while the lowest was found in wave-dominated deltas (median of 16.3 mg/L).

Long-term averaged RPAs also exhibited notable differences across the six continents. Large RPAs were found in South America

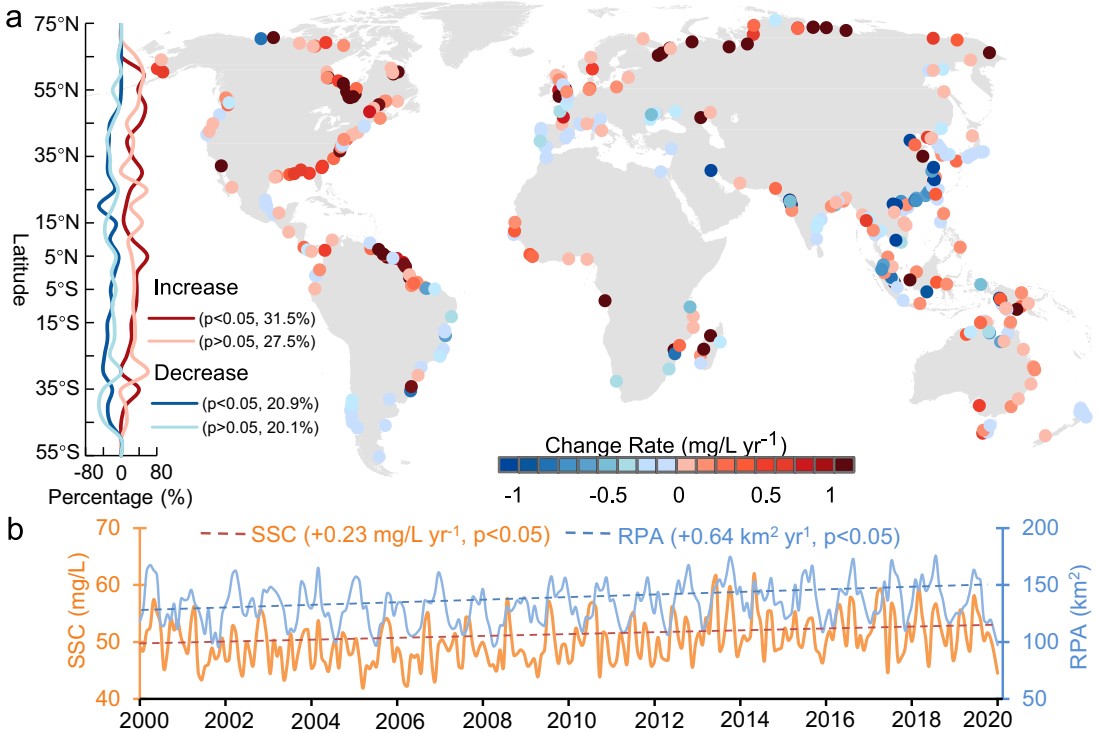

**Fig. 2 | Trends of coastal suspended sediment concentration (SSC) and river plume area (RPA) between 2000 and 2020. a** Spatial patterns of the SSC trends (Mann–Kendall test) in 349 deltas. The latitudinal profiles exhibit the percentages of deltas with significant ($p < 0.05$) and insignificant SSC change trends (increase or decrease). **b** Variations in monthly mean SSC and RPA, along with their long-term change trends at global scales. The orange and blue dash lines represent the long-term trends of SSC and RPA, respectively. These trends were derived based on long-term monthly anomaly SSC and RPA, which were estimated as the difference between the monthly mean SSC and RPA and their long-term average for that month (see Methods). The Sen's slope and $p$ value obtained from the Mann–Kendall test are annotated.

(median of 54.8 km²) (Supplementary Fig. 4a). However, North America exhibited the largest RPAs, with a median value of 78.8 km², despite having relatively low SSC values (Fig. 1 and Supplementary Fig. 4a). This can be attributed to certain deltas within North America that have large RPAs but low SSC values, such as the Saguenay (mean SSC: 8.7 mg/L; mean RPA: 151.5 km²) and Mississippi (mean SSC: 43.1 mg/L; mean RPA: 750.9 km²). Additionally, the median RPA of Arctic deltas is over two times that of non-Arctic deltas (Supplementary Fig. 4c). Heterogeneities were also observed across different delta morphologies, as both the river-dominated (median of 55.8 km²) and tide-dominated (median of 52.3 km²) deltas exhibited high RPAs, while the lowest RPA was found in wave-dominated deltas (median of 37.7 km²) (Supplementary Fig. 4b).

**Long-term trends in coastal SSC near global river deltas**

Global deltas experienced an increase of +0.46% yr⁻¹ (or +0.23 mg/L yr⁻¹, $p < 0.05$, Mann–Kendall test) in coastal SSC between 2000 and 2020 (Fig. 2b). 59% (206/349) of deltas had their SSC increases over the past two decades, with approximately twice as many deltas showing a significant ($p < 0.05$) increase compared to those experiencing a significant decrease in SSC (Fig. 2a). Increasing SSC is widespread across all six continents (Supplementary Fig. 5). Africa and North America exhibited the most significant increases in SSC, with a rate of +0.41 and +0.32 mg/L yr⁻¹, respectively. In contrast, deltas in Asia experienced a decline in SSC, with a mean decrease of −0.1 mg/L yr⁻¹, despite pronounced SSC increases observed in high-latitude regions (latitude >45°N) in Northern Asia (Fig. 2a). This decline has been attributed to recent dam constructions and sediment extractions in many rivers (e.g., Yangtze and Mekong) between 2000 and 2020[21,23]. Additionally, deltas along the east coasts of North America, where dams are older, exhibited prominent increases in SSC, while some deltas along the

southeast coasts of South America, such as Jequitinhonha (−0.59 mg/L yr⁻¹) and San Francisco (−0.34 mg/L yr⁻¹), experienced noticeable declines. Arctic deltas exhibited prominent increases in SSC, with a mean rate of +1.29 mg/L yr⁻¹, which is 18.4 times that of non-Arctic deltas (Supplementary Fig. 6a). Among three delta morphologies, the most pronounced increase was found in river-dominated deltas, with 62.0% of these deltas displaying an increasing trend and having a mean rate of +0.36 mg/L yr⁻¹ (Supplementary Fig. 6c).

We find that RPA also increases at +0.48% yr⁻¹ (+0.64 km² yr⁻¹, $p < 0.05$, Mann–Kendall test) globally (Fig. 2b). The number of deltas with a significant ($p < 0.05$) increasing trend in RPA was 78.2% greater than those with the opposite trend (Supplementary Fig. 7). Increases in RPA were also observed across different continents, with the most pronounced increases found in North America (+1.94 km² yr⁻¹) and Oceania (+0.72 km² yr⁻¹) deltas, such as the Mississippi (+11.9 km² yr⁻¹) and the Ord (+12.4 km² yr⁻¹) (Supplementary Figs. 5 and 7). However, the RPA in Asia exhibited an insignificant ($p > 0.05$) decreasing trend, likely due to the evident decline in many large rivers, such as the Yangtze (−64.2 km² yr⁻¹) and the Mekong (−2.8 km² yr⁻¹) (Supplementary Figs. 5 and 7). Similar to SSC, a considerable increase in RPA was observed in Arctic deltas, with a mean rate of +3.94 km² yr⁻¹, which is far more than that in non-Arctic deltas (mean rate of +0.23 km² yr⁻¹) (Supplementary Fig. 6b). Meanwhile, over 62% of both river-dominated and tide-dominated deltas exhibited an increasing trend in RPA, with a mean rate of over +0.8 km² yr⁻¹, more than double that of wave-dominated deltas (Supplementary Fig. 6d).

Within the latitudinal range of 30°N to 60°N, the number of deltas with a significant ($p < 0.05$) increase in RPA surpassed the numbers exhibiting other trends (e.g., insignificant increase/decrease and significant decrease) (Supplementary Fig. 7). However, such latitudinal patterns in SSC changes are not evident (Fig. 2a). Moreover, ~60% of

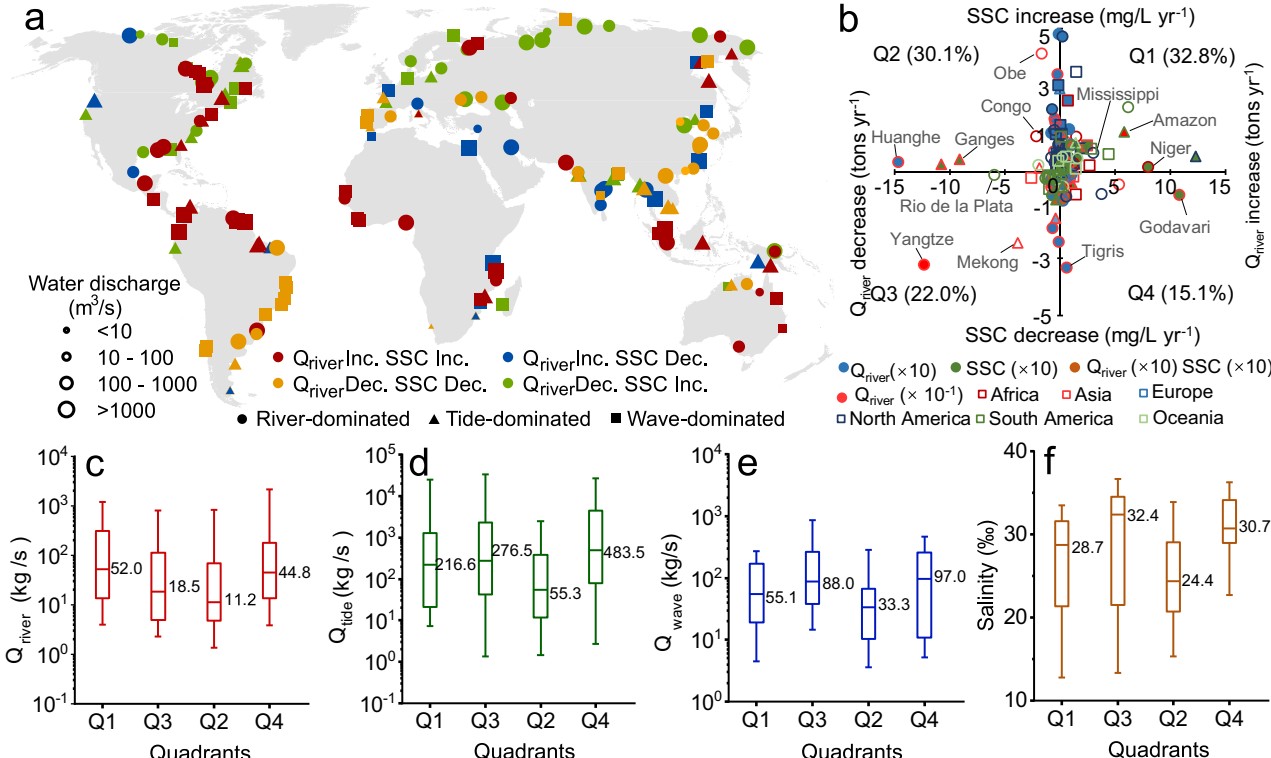

**Fig. 3 | Change trends of $Q_{river}$ and suspended sediment concentration (SSC) between 2000 and 2020 for 186 deltas. a** Spatial patterns of the trends (increase or decrease) in $Q_{river}$ and SSC. The different colors represent different combinations of $Q_{river}$ and SSC trends (Inc.: increase; Dec.: decrease). **b** Scatterplot of change rates (Sen's slopes from the Mann–Kendall test) in $Q_{river}$ and SSC for 186 deltas. Each quadrant (Q1, Q2, Q3, and Q4) represents a combination of SSC and $Q_{river}$ trends. The numbers in parentheses represent the percentages of deltas. The circle, triangle, and square represent river-, tide-, and wave-dominated deltas, and different colors represent different continents. The circles filled with orange represent that both the change rates of SSC and $Q_{river}$ are magnified 10 times, respectively. The circles filled with blue and green indicate a magnification of 10 times separately in only the change rates of SSC or $Q_{river}$, and the circles filled with red indicate a tenfold decrease in $Q_{river}$. The gray lines point out several representative deltas. **c–f** Box plots of long-term mean $Q_{river}$, $Q_{tide}$, $Q_{wave}$, and salinity in different quadrants in **b**. The box plot shows the distributions of 10%, 25%, 50%, 75%, and 90% values.

deltas with a latitude of around 5°N exhibited a significant increase in SSC, while such an increase was not notable in RPA. These trend disparities are mainly due to the complexity of the spatiotemporal response of sediment plumes to SSC variations under the influences of tide and wave forces. In fact, high sediment concentrations do not necessarily imply larger RPAs due to the intricate movements of sediment under the integrated effects of river flow, salinity, and tide and wave forces (Supplementary Fig. 3m–p)[24,25].

### Response of coastal SSC near delta to river sediment supply change

To understand the potential response of SSC to river sediment supply ($Q_{river}$), we compared the $Q_{river}$ change trends collected from ref. 6 with SSC variations for 186 deltas. Among these deltas, 47.9% have an increasing trend in $Q_{river}$ during 2000–2020 (Fig. 3). We observed consistent variations between $Q_{river}$ and SSC in 54.8% of examined deltas (Fig. 3). Specifically, among these 54.8% of deltas, 32.8% are gaining SSC as well as $Q_{river}$. Many of these deltas are large river- and tide-dominated deltas, such as the Amazon and the Mississippi (Fig. 3b). Consequently, these consistent increments may be attributed to the large river inputs and tidal forces, which then outweigh other controls on SSC (Fig. 3c, d) (see also in refs. 26,27). Meanwhile, 22% of the deltas experience a redcution in both SSC and $Q_{river}$, primarily located in South Asia and the southeast of South America (Fig. 3a, b). These deltas exhibit lower $Q_{river}$ and higher $Q_{wave}$ and salinity (Fig. 3c, d).

In contrast, we observed opposite trends between $Q_{river}$ and SSC in 45.2% of the deltas (Fig. 3b). Among these, 30.1% of the deltas

exhibited an increase in SSC while $Q_{river}$ declined, and 15.1% of the deltas showed a decline in SSC while $Q_{river}$ increased. Many of these deltas had lower $Q_{river}$ and weak tidal forces, or higher salinity and strong wave energy (Fig. 3c–f). These findings highlight the complexity of coastal sediment in response to $Q_{river}$. Other controls, such as wind, wave, or tidal currents, as well as salinity might obscure the response of SSC to $Q_{river}$[16,28,29]. Furthermore, anthropogenic activities may also contribute to opposite changes between $Q_{river}$ and SSC. For example, channel deepening and dredging activities have resulted in high SSC in the Ems delta, despite its small and decreasing $Q_{river}$[30].

### Controls on coastal SSC

Seasonal variations in coastal SSC typically correspond well to fluctuations in river sediment supply ($Q_{river}$, Pearson correlation coefficient: $R = 0.76$, $p < 0.05$), tidal sediment discharge in- and out of deltaic channels ($Q_{tide}$, $R = 0.71$, $p < 0.05$), and wave sediment movement ($Q_{wave}$, $R = 0.94$, $p < 0.05$) for individual deltas. Seasons with high river sediment discharge, spring tide, or strong wave energy often exhibit higher SSCs (Supplementary Fig. 8). However, such relationships do not necessarily extend to the long-term trends observed across deltas. Deltas with higher average wave sediment fluxes, for example, tend to have lower SSCs ($R = -0.40$, $p < 0.05$, Fig. 4c), while large tidal sediment movement ($R = 0.31$, $p < 0.05$) and high river sediment discharge ($R = 0.23$, $p < 0.05$) often lead to higher SSC (Fig. 4a, b). The deviation between individual seasonal and interdelta long-term relationships signifies the influence of coastal morphology on coastal SSC. This is evident from the fact that wavedominated deltas showed the lowest median SSC compared to river-

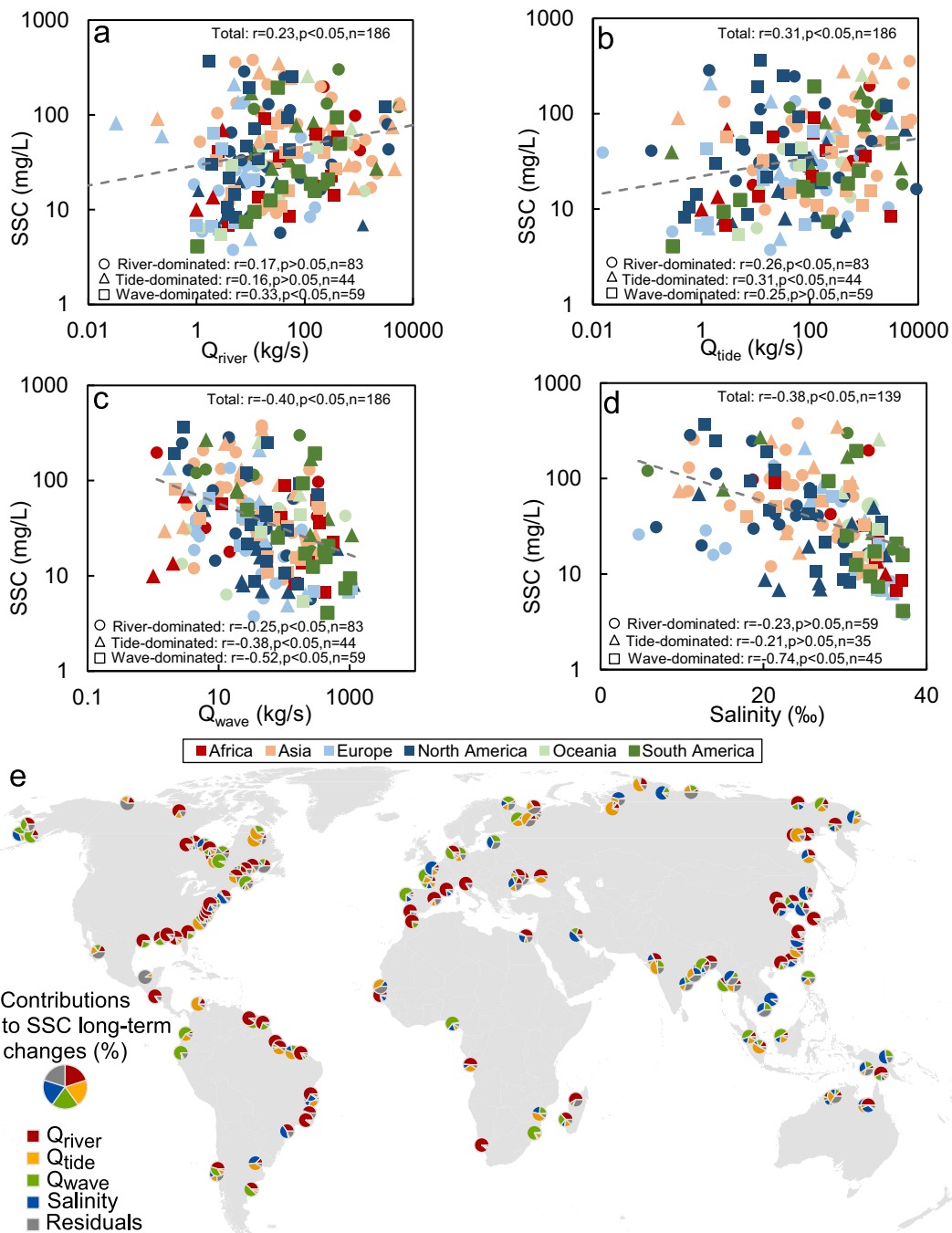

**Fig. 4 | Relationships between long-term mean suspended sediment concentration (SSC), $Q_{river}$, $Q_{tide}$, $Q_{wave}$, and salinity. a–d** The relationships between SSC and $Q_{river}$ (**a**), $Q_{tide}$ (**b**), $Q_{wave}$ (**c**), and salinity (**d**). The number of deltas (*n*), correlation coefficients (*r*), and *p* values were annotated. Out of our 349 deltas, the $Q_{river}$ is only available for 186 deltas and salinity is available for 139 deltas, thereby the relationship analyses between $Q_{river}$, $Q_{tide}$, $Q_{wave}$, salinity, and SSC were conducted for these deltas. The correlation coefficients are obtained based on the logarithm-transformed SSC and different factors. **e** Contributions of four drivers ($Q_{river}$, $Q_{tide}$, $Q_{wave}$, and salinity) on long-term SSC changes in 139 river deltas. The contributions (see Supplementary Table 3) of four drivers in each delta are presented through pie charts. The pie charts consist of five colors, representing the four drivers and residuals, with the size of each slice indicating the respective proportion.

dominated and tide-dominated deltas (Fig. 1). Moreover, wave-dominated deltas are more likely to have a decreasing SSC compared to river- and tide-dominated deltas (Supplementary Fig. 6c). This discrepancy between seasonal intra-delta and long-term inter-delta SSC dynamics suggests that wave-dominated deltas, characterized by the absence of distributary networks and sandy shorelines[31] (Supplementary Fig. 2a–f), do not efficiently retain coastal sediment.

Waves increase SSCs temporarily, but the long-term effect of higher wave dominance tends to lead to a decrease in SSC. In tide- and river-dominated deltas, on the other hand, seasonal and long-term SSCs change in the same pattern, showing their capacity to retain coastal sediments nearshore for extended periods.

In addition, we also found that nearshore salinity has an inverse relationship with coastal SSC. The long-term, high salinity exhibited a

significantly negative correlation with delta-averaged SSC ($R = -0.38$, $p < 0.05$, Fig. 4d). Meanwhile, a significant negative correlation ($R = -0.61$, $p < 0.05$) was observed between seasonal SSC and salinity for individual deltas, indicating that low SSC is greatly correlated to high salinity (Supplementary Fig. 8). These findings align with previous studies that have suggested an increase in coastal SSC (or turbidity) with a decrease in salinity[32,33], likely associated with high freshwater river discharge events that carry sediments, or the effects of salinity on sediment flocculation and settling[34].

To further comprehend the impacts of the four drivers ($Q_{river}$, $Q_{tide}$, $Q_{wave}$, and salinity) on SSC variations across different deltas, we assessed the contributions of these drivers to SSC long-term changes in 139 deltas (Fig. 4e, Methods). We observed that changes in $Q_{river}$, $Q_{tide}$, $Q_{wave}$, and salinity significantly ($p < 0.05$) accounted for the SSC changes in 36%, 13%, 9%, and 7% of the 139 deltas, respectively (Supplementary Table 3). On average, the combined influence of these four drivers explained $84.3 \pm 14.2\%$ of the variations in SSC across the 139 deltas. Increases in $Q_{river}$ and $Q_{tide}$ along the Atlantic coasts of North and South America contributed the most to SSC increases there, whereas their decreases, coupled with increases in $Q_{wave}$ and salinity, likely led to declines in SSC in South Asia (Fig. 4e and Supplementary Fig. 9). Moreover, we found that $Q_{river}$, $Q_{tide}$, $Q_{wave}$, and salinity significantly ($p < 0.05$) contributed to the SSC changes in 33%, 2%, 12%, and 7% of Arctic deltas, respectively (Supplementary Table 3). Despite the slight decreases in $Q_{river}$ and $Q_{tide}$ in Arctic deltas, the SSC of many deltas in this region has increased (Fig. 2 and Supplementary Fig. 9). This might be attributed to the weaker effects from wave and salinity, which have both decreased (Supplementary Fig. 9). These findings underscore the intricate responses of coastal SSCs to changes beyond river sediment supply.

## Discussion

We developed global coastal surface SSC and RPA datasets based on satellite images from 2000 to 2010 for 349 deltas over the past two decades. Our results indicate that in over 59% of global deltas, both coastal SSC and RPA have increased from 2000 to 2020. Prominent gains in SSCs and RPAs were observed across all six continents, except Asia, where many rivers have been widely reported to have an evident decline in river sediment flux to the ocean due to extensive dam constructions[7,35].

We did not observe covariation between SSC and RPA, possibly due to limitations in satellite observations of water surface SSC and RPA. In many coastal environments, especially those governed by estuarine and wave dynamics, surface SSC can be highly localized[36–38]. In our study, we found significant relationships between SSC and $Q_{river}$, $Q_{tide}$, $Q_{wave}$, and salinity (Fig. 4). Notably, $Q_{river}$ also exhibited a significant correlation with RPA (Supplementary Fig. 10a), as supported by other studies[39,40]. However, we did not observe a significant relationship between RPA and $Q_{tide}$ (Supplementary Fig. 10b), despite the documented impacts of tides on RPA[39]. This discrepancy may arise from strong three-dimensional effects in estuaries. For instance, tidal pumping and density-driven estuarine circulation could result in high sediment concentrations near the bed without generating a surface expression of tides on RPA[41–44]. In addition, waves may also confine river plumes nearshore (Supplementary Fig. 10c)[45]. Other factors, such as the strength and direction of the wind[46–48], could also modulate the horizontal extent of river plumes and obscure the relation between SSC magnitude and RPA. For instance, strong winds could result in significantly high SSC and an extensive RPA near the Yangtze estuary. In contrast, weak winds might lead to high SSC near the shore but a smaller RPA outside the river mouth, primarily due to the limited spread of turbid water[47]. Additionally, winds blowing towards the shore can lead to extremely high SSC near the coast and a constrained RPA in the Yangtze estuary, while offshore winds may result in relatively low SSC yet a broader RPA[48].

We observed that near certain deltas, coastal SSC are closely related to $Q_{river}$ variations, as has also been shown elsewhere[21,49]. However, not all deltas exhibited consistent changes between SSC and $Q_{river}$ (Fig. 3a, b), likely due to the complex influences from hydrodynamic forces (e.g., tide and wave), salinity, and human interventions (e.g., sediment dredging)[9,50–52]. For instance, although $Q_{river}$ has declined due to dam construction in the Mekong, there has been wave-driven enhancement of seasonal coastal SSC[20]. Additionally, river plume dynamics induced by river flow density may also complicate the relationship between $Q_{river}$ and SSC. Increases in $Q_{river}$ might densify the river flow, stimulating the development of hyperpycnal plumes without generating a surface expression in the SSC[53–55]. For example, in the Huanghe delta, ~80% of the river sediment on the delta front is deposited from hyperpycnal flows[56], making surface SSC signature hard to detect. However, satellite imagery is relatively easy to capture the distinct surface SSC signature for low-density hypopycnal plumes[53,57,58]. As such, the satellite observed variations in SSC from the water surface may have poor correlations with changes in river sediment supply (Fig. 3b).

The global SSC dynamics that we observe and their dependence on $Q_{river}$ offer valuable insights into sediment availability for coastal wetlands. Coastal SSC is a vital predictor when evaluating the resilience of coastal wetlands in the face of sea-level rise[19]. Previous research has raised concerns that the decline in $Q_{river}$ could potentially limit the availability of coastal SSC, thereby threatening the survival of wetlands[2]. To protect coastal wetlands, various studies have therefore proposed removing dams to enhance $Q_{river}$ and subsequently increase coastal SSC[59,60]. However, this may be less straightforward than previously thought. Our results indicate that not all $Q_{river}$ declines will lead to SSC loss, at least within the timeframe of our analysis (Fig. 4b). In addition, we observed a prominent increase in SSCs over the past two decades. This could enhance wetland resilience to future sea-level rise, and suggest that wetland vulnerability might be overestimated in previous studies[1].

Nonetheless, the response of coastal wetlands to SSC change appears to be complex. Through a comparison of changes between wetland areas (from ref. 61) and coastal SSC across 180 matched deltas, we found that nearly one-third of wetlands experienced area loss despite an increase in coastal SSC (Supplementary Fig. 11). This phenomenon can likely be attributed to human activities, such as coastal development[59] and land conversions[62], as well as natural influences like land subsidence and accelerated sea-level rise[63]. Therefore, although the link between wetland accretion and coastal SSC is well-established[64], it does not necessarily imply that increases in SSC will lead to wetland gains. The complex connections between global coastal SSC and wetlands necessitate a thorough analysis, particularly with the anticipated availability of more extensive data in the future.

Our SSC dataset could also serve as a valuable resource for the management and restoration of coastal environments. Increased coastal sediment can lead to delta area expansion, as large sedimentation creates new mudflats, which could provide more habitats for various creatures[65]. However, not all increased SSCs have positive effects on coastal ecosystems[66]. Elevated SSCs can affect underwater photosynthesis and the survival of aquatic organisms due to increased water turbidity and reduced underwater light visibility[67]. Furthermore, the increased sediment can exacerbate the deterioration of water quality and even promote algal blooms, as sediment is one of the significant carriers of land-based pollutants (such as pesticides, nutrients, and heavy metals)[68,69]. For example, the nutrients released from sediment, such as phosphorus and nitrogen, have enhanced the phytoplankton blooms in many lakes and coastal regions[70,71]. As such, whether the increased coastal SSC is beneficial to the coastal zone needs further local investigation.

In conclusion, our study contributes to understanding the spatiotemporal patterns of coastal SSC near deltas. We observed a global

increase in delta coastal SSC and RPA between 2000 and 2020. This increase is partially explained by increases in river sediment supply, but some deltas go against the grain. Waves, tides, salinity, and coastal morphology also modulate coastal SSCs and explain SSCs gains despite declines in river sediment supply (Fig. 4). We also found that not all coastal wetlands exhibited consistent fluctuations with SSC. Our results offer valuable insights into the present distribution and dynamics of coastal sediments near river deltas. The SSC dataset can be utilized to assess the deltaic sediment balance around coastal wetlands and evaluate the threat of sea-level rise. Furthermore, it could offer crucial information for the protection and restoration of coastal areas, especially those experiencing sediment deficits.

## Methods

### Data sources

The MODIS Terra and Aqua 8-day SR products, with a spatial resolution of 500 m, were utilized to estimate SSC. We used 0.31 million Terra images (MOD09A1) (2000–2020) and 0.27 million Aqua images (MYD09A1) (2002–2020), totaling 0.58 million composites. All products were atmospherically corrected using an atmospheric correction algorithm based on the 6S radiative transfer[72]. A Quality Control (QC) flag indicating the quality of each pixel was used to exclude the potential artifacts (such as atmospheric correction failure) from SSC estimation. In addition, MODIS Aqua daily SR product (MYD09GA), consistent with the field sampling dates, was used to build the SSC inversion algorithm. The MODIS SR products are all accessible on Google Earth Engine (GEE).

We used six field SSC datasets to calibrate and validate the SSC inversion algorithm, including the Pearl River estuary, the Yangtze River estuary, the Yellow and East China Seas (YECS), the SeaSWIR, the CoastColour Round Robin (CCRR), and the AquaSat. Details are presented in Supplementary Fig. 1a.

We conducted three cruise surveys in the Pearl River estuary from September 2018 to January 2020, comprising 64 sampled stations. The SSCs of these samples range from 0.16 to 137.5 mg/L. Three field surveys were arranged in the Yangtze River estuary in February and June 2011 and March 2013, with 99 water samples collected. The collected SSCs vary from 0.1 to 2068.8 mg/L. The YECS dataset includes 150 water samples measured from Spring and Autumn cruises covering the Yellow and the East China Seas in 2003, with SSCs varying from 0.5 to 1762.1 mg/L.

The SeaSWIR is a publicly accessible dataset[73], including 137 sediment samples collected from three turbid estuarine sites: the Gironde of France, the Río de La Plata of Argentina, and the Scheldt of Belgium. The SSCs in this dataset range from 48.4 to 1400.5 mg/L.

The CCRR, also an online free dataset[74], compiles diverse in situ SSC data measured from multiple coastal regions worldwide. This dataset was designed to calibrate and evaluate algorithm performance. The SSCs used in this research range from 0.17 to 506 mg/L.

The AquaSat assembled various in situ water constituent data encompassing inland and coastal waters across the USA. This compilation was sourced from two existing publicly available datasets, namely the Water Quality Portal and LAGOS-NE[75]. AquaSat contains over 400,000 estuary sediment samples. However, we only considered samples meeting two specific criteria: (1) collected after 2002 to align with MODIS Aqua data availability and (2) situated at open river mouths to mitigate potential disturbances from land adjacency effects. A total of 10,686 samples met these criteria, with SSCs ranging from 0.2 to 305.3 mg/L.

To validate the accuracy of our SSC inversion algorithm, we acquired the monthly mean 4 km SSC products generated from Sentinel-3 OLCI spanning 2016–2020. These products were collected from https://sentinels.copernicus.eu/web/sentinel/user-guides/sentinel-3-olci/product-types/level-2-water, which is derived using the inherent optical property—the backscattering coefficient.

The inversion formula is expressed as: $SSC = 1.06 \times B_{bp442.5}{}^{\wedge} 0.942$. Due to the coarse spatial resolution of Sentinel-3 OLCI, data are only available for 227 out of the 349 deltas studied. The long-term monthly dynamics and the long-term mean OLCI SSC were used to validate SSC inversion accuracy in this study.

The long-term mean river water discharge (in m³/s) data for 349 deltas were collected from ref. 5 to investigate the river water discharge on SSC variations. These mean water discharges were generated from river discharge stations.

We obtained the annual river sediment flux ($Q_{river}$) between 2000 and 2020, as well as the multi-year monthly mean $Q_{river}$, from a global fluvial sediment flux dataset created by ref. 6. Out of the 349 examined deltas, 186 deltas were successfully matched with ref. 6's dataset. The $Q_{river}$ dataset was generated using the river SSC derived from Landsat images and water discharge from matched river discharge stations. The Landsat-derived SSC mainly originates from pixels in the river channels, which may be several times larger than our coastal SSC. Such a significant discrepancy arises partly due to the size and flow rate of the river, and also because of the different areas estimated (e.g., river channels vs river coasts), which have different driving forces. The water discharge (measured in cubic meters per second, m³/s) represents the fluvial flow passing through the entire river cross-section. Note that the $Q_{river}$ employed in this study, derived from remote sensing surface SSC and water discharge, may be more applicable to the well-mixed macro-tidal estuaries. This is because, in meso- or micro-tidal estuaries, three-dimensional effects may significantly modulate river flow, introducing uncertainties in $Q_{river}$ analysis. In the future, to obtain a more accurate $Q_{river}$ in these systems, the combination of remote sensing and modeling could be considered as an alternative approach.

We collected the monthly tidal sediment flux ($Q_{tide}$) using the monthly mean tidal amplitude ($a$), angular velocity ($\omega$), channel slope ($s$), tidal efficiency coefficient ($k$), upstream channel depth ($du$), and the channel aspect ratio ($\beta$) of estuaries. The daily tidal amplitudes during 2000–2020 were extracted from OSU TPXO, and were aggregated into monthly means. Other coefficients were obtained from ref. 76. Similar to ref. 5, $Q_{tide}$ was evaluated as follows:

$$Q_{tide} = Q_{wtide} \times Q_{river} / Q_{wriver} \tag{1}$$

Where $Q_{river}$ is fluvial sediment flux (kg/s) and $Q_{wriver}$ denotes water discharge (m³/s), both of which are obtained from ref. 6.

We assessed the daily wave sediment flux ($Q_{wave}$) using the daily mean wave height ($h$) and wave period ($p$) from 2000 to 2009 extracted from NOAA WaveWatch III for each delta, following the method of ref. 5:

$$Q_{wave} = 2650 \times 0.6 \times k_1 \times h^{2.4} \times p^{0.2} \tag{2}$$

Where $k_1$ is an empirical constant, approximately equal to 0.06. Further details about $k_1$ are described in ref. 31. Finally, all daily $Q_{wave}$ values were aggregated into monthly means.

All the monthly $Q_{tide}$ and $Q_{wave}$ data were aggregated into annual means to track long-term changes and their contributions to SSC changes. As $Q_{wave}$ data are not available after 2010, we utilized the mean values from 2009 to represent conditions from 2010 to 2020 and conducted the trend and contribution analyses. Finally, these three sediment flux data were collected for the 186 matched deltas to investigate the relationships between hydrodynamic forces (river sediment discharge, tides, and waves) and changes in SSC.

We utilized the monthly salinity dataset from 2000 to 2020 to explore the relationships between salinity and coastal SSC changes. This dataset was acquired from the GLORYS12V1 product (https://data.marine.copernicus.eu/product/GLOBAL_MULTIYEAR_PHY_001_030/description), which is a reanalysis product with a spatial resolution of 1/

12° (~8 km). Due to the coarse resolution of this product, only 139 out of 349 examined deltas had available salinity data. All the monthly salinity data were aggregated into annual mean to track the long-term salinity changes and their contributions to SSC changes.

We obtained wetland change data for the 349 deltas from the global tidal wetland change dataset[61], available on the GEE platform, to investigate potential responses of wetlands to SSC changes. This dataset was derived from Landsat satellite images and offers insights into global wetland gain and loss extents within a 5-km buffer around intertidal ecosystems or along the coastal line from 1999 to 2019[61]. Among the 349 deltas examined, 180 deltas were matched with the wetland dataset.

We acquired the dataset on delta morphologies from ref. 5, which classified deltas into river-dominated, tide-dominated, and wave-dominated categories based on the dominant sediment flux shaping delta morphology. This data was utilized to assess the relationship between delta morphology and SSC change.

## Determination of studied river deltas

We examined 349 river deltas distributed across the globe, ranging from small to large deltas, with mean annual river water discharges from 1.3 to 138,650 m³/s, representing 64.2% of the global total water discharge[5]. These deltas were compiled from ref. 5, encompassing a diverse range of delta systems, including river-dominated, tide-dominated, and wave-dominated (Supplementary Fig. 2).

## SSC retrieval algorithm

To retrieve coastal SSC with a wide dynamic range (e.g., from clear to turbid) using satellite images, numerous algorithms have been developed over the past decades[77–80]. Nevertheless, most of these algorithms require a reflectance threshold from specific wavelengths as the blending boundaries, which varies among different research. Recently, several algorithms for estimating global coastal sediment have been proposed[18,22,81], enabling the quantification of coastal sediment at global levels. In this study, we conducted accuracy assessments for these algorithms based on in situ measured data and MODIS SR data to select the most accurate algorithm for SSC inversion.

Initially, we selected matchups between the daily MODIS Aqua SR data and in situ SSC data for the same day, excluding samples within three pixels of land-water boundaries or clouds to prevent land or cloud contaminations. This resulted in a total of 509 matchups. Subsequently, we used these matchups to conduct a recalibration (details see Supplementary Note 1) of several typical existing coastal SSC inversion algorithms, including the algorithm from ref. 77 (hereafter as Han_adapted), ref. 78 (hereafter as Feng_adapted), and ref. 22 (hereafter as Yu_adapted), and assessed their accuracy. We found that after calibrating algorithm parameters based on our in situ measured data and MODIS SR data, Han_adapted had the largest error (overall accuracy >51%), with RMSE exceeding 50% in both clear (<50 mg/L) and turbid water (≥50 mg/L) (Supplementary Table 1). Feng_adapted achieved a better overall accuracy of around 28%, with accuracies of 28.5% for clear water and 48.6% for turbid water. The model with the highest accuracy was Yu_adapted, with an overall accuracy of 24.9%, particularly performing well in clear water (RMSE = 24.9%), and 49.1% for turbid water (Supplementary Fig. 1 and Table 1). Considering the SSCs of the majority of deltas are not extremely high, in this study, we adopted the Yu_adapted algorithm (see below) to invert SSC from MODIS SR products.

$$SSC = \exp(0.859*(0.145*(R_{555}/R_{469}) + (5.167*(R_{645}/R_{555})$$
$$*(R_{645}/(R_{645}+R_{859})) + (7.244*(R_{859}/R_{555})$$
$$*(R_{859}/(R_{645}+R_{859}))))^{\wedge}0.990) \quad (3)$$

Where $R_{469}$, $R_{555}$, $R_{645}$, and $R_{859}$ represent the blue, green, red, and NIR reflectance from MODIS SR products.

To further validate the accuracy of Yu_adapted, we first compared the ranges of our MODIS-derived SSC to the previously reported SSC through a review of published literature. In total, 31 deltas across the globe were examined (Supplementary Table 2). We observed that our MODIS-derived SSC aligns well with the published SSC for most deltas, yet discrepancies also exist in some deltas. Considering the disagreements of the investigation time and satellite sensor selections, these disparities are considered acceptable. Additionally, we compared the multi-year average SSC between 500-m MODIS and 4-km Sentinel-3 OLCI, noting that the average SSC from MODIS compared to OLCI has a mean ratio of 1.41 ± 0.83 (Supplementary Fig. 1e). However, when SSC exceeds 100 mg/L, the average SSC from MODIS is noticeably higher than that from OLCI. Given the differences in spatial resolution, we believe such discrepancies are acceptable. Furthermore, we separately compared the temporal variations in monthly mean SSC from 2016 to 2020 for several typical estuaries, ranging from clear to turbid. We observed that the trends based on MODIS and OLCI are very similar (Supplementary Fig. 12). Consequently, we applied the Yu_adpated algorithm to MODIS Aqua and Terra 8-day SR products to retrieve coastal SSC for the 349 deltas. The MODIS Terra has been noted for its radiometric degradation and calibration errors, often deemed unreliable in the ocean color community[82]. However, we observed comparable magnitude and dynamics of SSC from Terra and Aqua across clear to turbid waters (Supplementary Fig. 13), which has also been documented in previous research[83]. Therefore, we adopted Terra data to complement Aqua observation in this research.

We acknowledge that ideally, the MODIS remote sensing reflectance (Rrs), generated using an atmospheric correction approach tailored for ocean color application[84], should be utilized to derive SSC. Since the MODIS SR were produced using a land-based atmospheric correction algorithm[72], which does not correct for skylight reflection at the air-water interface (e.g., Fresnel reflection), potentially introducing errors in clear water body inversion. However, a high spatial resolution (i.e., <1000 km) Rrs product is not readily available globally, which limits monitoring capabilities for small estuaries or bays. Indeed, the MODIS SR has been proven to have good agreement with Rrs in both spatial and temporal patterns in turbid inland or coastal water[85]. Moreover, the skylight reflection problem could be mitigated to some extent through band ratio methods[24,86]. More importantly, the MODIS SR product (with a resolution of 500 m) has global coverage and can be freely accessed through GEE, enabling general users to perform global applications. To date, the MODIS SR has been widely used in inland and coastal water quality monitoring[83,87,88]. In the future, with continued advancements in data processing and storage capabilities, we believe it will be feasible to estimate global SSC based on Rrs derived from ocean color atmospheric correction.

## Determination of the studied region and RPA of the river delta

To determine the final study region for 349 deltas, a 70% frequency with SSC over a specific SSC threshold was adopted to delineate the region boundary (Supplementary Fig. 14). The detailed methodology is outlined as follows: (1) A point near the river mouth was designated, and a 5-km buffer around this point was computed. Subsequently, median SSC values within the 5-km buffer in each monthly mean SSC image during 2000–2020 (totaling 251 months) were collected. (2) These 251 median SSC values were arranged in ascending order, and the 5th percentile (5%) of these ascending SSC was selected as the SSC threshold (SSC_threshold). Then the frequency of each pixel with SSC over SSC_threshold in the 251 months was counted, and pixels with a frequency exceeding 70% of 251 (e.g., 251 × 70% = 176) were chosen to form the final study region.

To assess the impact of the study region determined by using different frequency percentage thresholds (e.g., 70%) on the average SSC, we performed a sensitivity test. Specifically, we calculated the mean SSC determined by using low-frequency ($S_{low}$) and

high-frequency ($S_{high}$) thresholds, respectively. Subsequently, we calculated a relative difference (RD) between $S_{low}$ and $S_{high}$ to illustrate the potential impacts arising from the use of different frequency thresholds (for example, $RD_{65\%} = (S_{70\%} - S_{65\%})/S_{65\%}$). We found that the RD remains at 5% when employing a frequency percentage from 50% to 70%, yet increases to over 10% when using a frequency threshold exceeding 70%. Consequently, we opted for a threshold of 70%.

The monthly RPA was delineated for the aforementioned final study region using the monthly mean SSC and the specific SSC threshold for each delta. We identified the pixels with a monthly mean SSC exceeding the $SSC_{threshold}$ (as described above), and the maximum extent formed by all these pixels was regarded as the RPA for that month (Supplementary Fig. 3).

Given the potential for incomplete data coverage in monthly synthesis from Terra and Aqua, we assert that the monthly SSC and RPA are reliable only when the valid data in monthly images exceeds 50% (i.e., a ratio of valid pixels to the total pixels in study region). Through sensitivity analysis, we found that when the valid ratio threshold varies from 50% to 70%, less than 2% of deltaic SSC trends alter. However, when the threshold exceeds 80%, over 5% of deltaic trends change, likely due to a notable reduction in the data volume used in trend analysis. We noted that, for most months between 2000 and 2020, valid coverage for the majority of deltas exceeded 50%. Hence, the 50% threshold was adopted. Nonetheless, certain deltas may not meet the 50% threshold in specific months. In such instances, we mitigated data gaps by averaging data from preceding and subsequent months or from the same month in adjacent years. It's important to mention that some high-latitude estuaries may lack data during winter and spring seasons. In such instances, our statistical analysis relies solely on available data for certain quarters.

### Delta wetland area changes
We gathered data on wetland gain and loss extents near deltas from the wetland dataset and computed the net wetland area change (in $km^2$) for each delta by taking the difference between increased and decreased area.

### Statistical analysis
The SSC and RPA monthly mean anomaly, determined as the differences between the monthly mean SSC and RPA and their corresponding long-term average for that specific month, were used to examine the long-term change trend in these two parameters over the past two decades. This anomaly processing is primarily used to remove the seasonal interference in long-time series data, thereby achieving precise monitoring of long-term changes[89]. We employed Sen's slope (obtained from the Mann–Kendall test) of the long-term monthly mean anomaly, multiplied by 12 (representing 12 months a year), as the mean yearly change rate to evaluate the SSC and RPA variations for each delta, and the associated $P$ value ($p$) was used to assess whether the changing trends were statistically significant ($p < 0.05$). The monthly anomaly SSC and RPA at global or continental scales were generated by using the average monthly SSC and RPA from 349 deltas or from deltas located on different continents.

Furthermore, the Sen's slope (Mann–Kendall test) of the annual mean $Q_{river}$, $Q_{tide}$, $Q_{wave}$, and salinity was adopted to indicate the long-term changes in $Q_{river}$, $Q_{tide}$, $Q_{wave}$, and salinity, with associated $p$ value used to assess the significance of the change trends. The trend analysis using Mann–Kendall was conducted in MATLAB 2021.

Correlation analyses were conducted to assess the relationships between the annual mean SSC and four drivers ($Q_{river}$, $Q_{tide}$, $Q_{wave}$, and salinity). Subsequently, a multiple general linear model (GLM) regression analysis[90] was performed to quantify the contributions of these four drivers to SSC changes over the past two decades. The relative contribution of each factor was determined by calculating the ratio of

the mean sum of squares (MeanSq) of the specific driver and the total MeanSq (Supplementary Table 3). The $p$ value (e.g., $p < 0.05$) was estimated to examine whether the correlation coefficients and contributions were statistically significant. Due to the data unavailability for all four drivers, only a total of 139 examined deltas were included in the GLM analysis. Both correlation relationship and GLM analysis were conducted using R 3.3.0.

## Data availability
The entire MODIS-derived global deltaic coastal SSC data in this study have been deposited in the Figshare database under accession code https://figshare.com/s/153cf61cd73819cb2e30.

## Code availability
Code to reproduce the findings is available on https://codeocean.com/capsule/6854747/tree.

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

## Acknowledgements

We thank the NASA LP DAAC for providing MODIS SR data and GEE for providing image processing resources. We also thank Dr. Murray for providing the wetland area change data and Dr. Dethier for providing the river sediment flux data. L.F., X.H., and F.S. were supported by the National Natural Science Foundation of China (Nos: 42321004, 42301392, and 42271348). X.H. was supported by the Fundamental Research Funds for the Central Universities, Sun Yat-Sen University (NO: 23qnpy08). J.N. was supported by the National Science Foundation (EAR-GLD-1810855) and the Dutch Research Council (VENI.192.123).

## Author contributions

X.H.: methodology, data processing and analyses, and writing; L.F. and J.N.: conceptualization, methodology, supervision, and writing; D.X.: analyses and writing; F.S.: in situ data acquisition and writing. All authors participated in interpreting the results and refining the manuscript.

## Competing interests

The authors declare no competing interests.
