## [Peer Review File · Nature Communications]

Sustained increase in suspended sediments near global river deltas over the past two decadesREVIEWER COMMENTS

Reviewer #1 (Remarks to the Author):

The subject is of major interest to readers and deserves to be published scientifically. It is, however, a very delicate subject, given the large number of factors influencing the quantity and distribution of suspended sediment at the mouths of major rivers (terrestrial, marine and atmospheric factors). The trends presented in the article need to be verified (see full commentary below), but the data presented in the article are nonetheless very interesting. However, the article has one major flaw: the analysis is incomplete, it does not delve deeply enough into the processes and mechanisms involved, and the comments on the results refer to and rely too much on presuppositions and general assessments. SSC and RPA have only secondary relationships and no reason to co-evolve, especially between sites. RPA is clearly conditioned by river discharge, which is itself partly constrained by the water regulation of dams, and is very sensitive to wind. SSC depends on the river water flow, the sediment supply (Q_{river}) conditioned by the erosion of the catchment, its size and slope profile (several parameters that are integrated into the Sediment Delivery Ratio), rainfall, climate and management (dams, dredging and sand mining in particular, but also land use change). A small sloping watershed on a high island in Oceania contributes much more sediment to the sea than a larger watershed with settling and storage plains, for the same flow rate. SSC is also influenced at the coast by estuarine resuspension, stratification (which results from the ratio between buoyancy and the mixing energy provided by the tide) and resuspension by waves. The upstream estuarine transport of sediment resulting from tidal pumping, which is particularly noticeable in the dry season, brings particles from the sea into the estuary, whose fate is partially mixed with the particles brought in by the rivers. Another important factor is that the impact of dams takes time to become established before a river settles into a new hydro-sedimentary regime. While catchment areas that have had dams for decades, such as those in the United States and Europe, have had no impact over the last twenty years because they are much older, recently installed dams in Asian catchment areas have had a major impact in reducing inputs over the last twenty years.

When the article mentions "the effect of sediment supply on SSC more complicated than previously assumed" (L. 24-25) or mentions "inconsistent trends" (L. 25, L. 145), the observation seems to be based on the authors' impression. To help readers find their way around, this is the perfect opportunity to list the main factors that may explain why SSC and RPA are changing either in the same direction (both increasing or both decreasing) or in opposite directions. For a given plume, if the wind comes from the sea and sticks the plume to the coast, RPA will be smaller and SSC larger than with an onshore wind. Opposite or inverse trends are not inconsistent ones.

In addition, the article probably contains an error in figure 2b (also Extended Data Fig. 3 & 4) that must be corrected for a later version: the units cannot be mg/L on the left, with positive or negative values (what is a SSC value < 0 ?), or areas on the right in km², positive or negative. The figure caption indicates "trends between 2000 and 2020": are the units mg L⁻¹ year⁻¹ or mg L⁻¹ decade⁻¹ or ...? Same concern for km² year⁻¹ or decade⁻¹. So the variables are not SSC and RPA but [SSC/Time] and [area/Time] or equivalent. This needs to be corrected. Then I wonder about the values given in the text for the resulting

slopes. If the units of the parameters presented are already a quantity per unit of time (e.g. SSC/T), calculating the trend amounts is thus calculating a second derivative? This point really needs to be clarified.

The section comparing SSC and the "river sediment supply" (L. 186) clearly illustrates that SSC is far from representing the sediment supply. Global values are given without comment. The discussion would benefit from an introduction to the 3D processes occurring at river mouths. For example: the sediment supply integrates inputs from the surface to the bottom and the three-dimensional effects in the estuary and in the plumes are major. While there is an export to the sea at the surface, there is often an import of sediment to the land in the salt wedge generated by tidal pumping, when there is any. It should also be noted that the Q_{river} values used come from an article submitted but not yet published (Sun et al., Nature climate change). The methodology raises questions: how did these authors include vertical profiles (stratified or not) in their estimate of Q_{river} based solely on surface suspended sediment data from Landsat data (L. 454)? Did you try to compare their SSC from Landsat with yours from MODIS? Until this article is published, it is impossible to assess the methodology used in the present version. The same applies to Q_{tide} , whose calculation formula uses Q_{river} collected from Sun et al (submitted).

A separate analysis of trends by delta morphology type could be instructive. The fine fraction dominates at the surface in SSCs, and is more likely to be deposited (and resuspended) in tide-dominated deltas (which are generally dominated by fine sediments) than in wave-dominated deltas (where coarse sediments dominate, and where fine sediments are therefore not deposited because of excess energy). It should be noted, however, that as a result of changes in river inputs, it is quite possible that some river-dominated deltas may have recently become wave- or tide-dominated, due to the strong decrease in river sediment supply. This point could also be addressed.

The discussion introduces generalities that are either unnecessary or too vague:

- L.230-231: "Our analysis... can provide valuable insight for predicting the future of coastal wetlands"). Which kind of prediction can be done using the article?
- L. 233-234, "the common concern is that Q_{river} decline would likely constrain SSC availability": please refer to the poor correlation between Q_{river} and SSC (Fig. 3a).
- L. 270-273 "our results ... can aid in forecasting the fate of ...". How so?

These general statements are inaccurate, unsubstantiated, unnecessary, and confusing. The results of the study, if well presented, will suffice to support a very informative article from a scientific point of view and on the dynamics underway in front of river mouths without leading to such unsupported generalities.

Overall, the article contains some very interesting information, which needs to be corrected (Fig. 2b). The explanations and comments are not explicit enough, ignore too much the diversity of catchment areas and 3D processes at river mouths, and are sometimes based on unsubstantiated assumptions. First of all, replace "inconsistent trends" with "opposing trends", and you will be in a more suitable position to detail the hidden mechanisms and take account of their wide diversity.

This article cannot be published as it stands, but it deserves some in-depth work before publication.

Sylvain Ouillon

Reviewer #2 (Remarks to the Author):

Summary and overall appreciation

This study uses 20 years of satellite data (MODIS surface reflectance, monthly composites at 500 m spatial resolution, from 2000 to 2020) to reconstitute the evolution of suspended sediment concentration (SSC) and river sediment plume area (RPA) over 349 deltas worldwide. Results are reported globally then distinguishing the 6 continents. Temporal trends are computed and related to the respective influence of tides, waves and delta morphologies. The results obtained are rather unexpected: an increase of coastal SSC is revealed despite a recognized “decline in global river sediment supply to the coast over the past few decades.”

The study is of high interest: coastal environments, notably those influenced by river inputs, are highly impacted by climate change effects and human activities and must be carefully monitored. Other studies (e.g., cited literature and references therein) already addressed this topic notably using ocean color satellite data, but usually at regional scales or globally over a shorter time period. Results and analyses in the present studies provide some new insights but also rise questions. The amount of work (e.g., number of satellite images processed and analyzed) is impressive, however several methodologies issues should be clarified to fully appreciate the results and conclusions of the study.

Detailed questions and comments are provided hereafter to improve the study and current version of the manuscript.

“decline in global river sediment supply to the coast over the past few decades”

This is not obvious and strongly regional (Li et al. 2020) e.g., with the north-south divergence and shift from Asia to South America stated by Dethier et al. (2022)

Main, 1st paragraph: what about coastal erosion??

Methods:

Why not using the Han et al. (2016) or Wei et al. (2021) global SPM algorithm?

Han B, et al. Development of a Semi-Analytical Algorithm for the Retrieval of Suspended Particulate Matter from Remote Sensing over Clear to Very Turbid Waters. Remote Sensing. 2016; 8(3):211.

<https://doi.org/10.3390/rs8030211>

More importantly would your results be similar or different if using their SPM algorithm? In other words, are your results (temporal trends) dependent of the SPM estimation method you used? This is a crucial point to be clarified. Can we define one reference valid method to estimate globally SPM in coastal waters using ocean color satellite imagery?

Different studies relying on different methodologies provide different results...we need to define the best reliable method

Your method is based on monthly composites of coastal SSC, is the monthly temporal resolution justified?

Your first objective is not really original...the second is more ambitious

Mapping global coastal SSC

Authors should clarify if their study focuses on SSC in river deltas or coastal zones; I understand it is focused on river deltas and does not really extend to coastal zones.

Can temporal trends be accurately detected combining 2 satellite sensors MODIS-T and MODIS-A? Would using only one sensor be more reliable for trend detection?

Methodology: I would actually suggest the authors to review existing methods to map globally coastal SSC, identify once for all the best method, i.e., the one to use to detect changes at global and regional scales along the last two decades. This would be useful to start the study.

You assume your SSC retrieval algorithm is accurate according to Extended Data Fig. 1: for me this figure clearly shows your algorithm:

> is not at all accurate for $SSC < 10$ mg/L (high scatter and overestimation in Fig. 1A)

> is not calibrated with enough in situ data for $SSC > 50$ mg/L

> is not validated with enough in situ data for $SSC > 100$ mg/L

These are 3 potential serious limits in your algorithm and the main reason why you must explain why using this algorithm instead of, e.g., the one developed by Wei et al. (2021) global SPM algorithm? And how does the choice of the algorithm impact on detected trends? Or you should limit your analysis to river deltas where SSC ranges from 10 to 50 mg/L?

MODIS surface reflectance (SR) products: despite the study by Feng et al. (2018), it is still questionable to use such a product for low turbidity deltas (e.g., the Ebro, Spain) as the atmospheric correction processing was developed for land surfaces.

Fig. 1: In addition to the classification of river deltas over the 6 continents, I would suggest to consider separately Arctic rivers as presumably more impacted by climate change effects.

Long-term trends in global coastal SSC

Global increase of +3.7% per decade, i.e., +4.7 mg/l per decade: does the method used (and uncertainties associated to satellite-derived SSC) allow detecting such small variations?

Statistical methods exist to detect not only one but several trends over a selected time period, e.g., Muggeo (2003): estimating regression models with unknown breakpoints. *Stat. Med.* 22, 3055–3071.

The authors should better explain/justify the method they adopted for the statistical analysis of deltaic coastal changes and impacts on resulting trends over two decades. For example, I can clearly see several periods in the time series generated for each continent in Extended Data Fig. 3.

Fig. 2: the figure legends and/or caption should be changed as we do not expect negative values for SSC (mg/L) no RPA (km²).

Controls on coastal SSC

Fig. 3: data points could be colored to distinguish the different regions of interest, i.e., the 6 different continents.

Could the authors also present/analyze the time series of Q_{river} , Q_{tide} , Q_{wave} and salinity globally and per continent?

Color Key

Black: Reviewers' comments (**Numbered**, e.g., **1.1**)

Blue: Author's response (**Numbered**, e.g., **R1.1**) and text from the previous manuscript.

~~Red~~: Text removed from the previous manuscript

Gold: new text integrated into the revised manuscript

Note that the line numbers mentioned here refer to the Word version with tracked changes (e.g., Lines 23).

REVIEWER COMMENTS

Reviewer #1 (Remarks to the Author):

The subject is of major interest to readers and deserves to be published scientifically.

It is, however, a very delicate subject, given the large number of factors influencing the quantity and distribution of suspended sediment at the mouths of major rivers (terrestrial, marine and atmospheric factors). The trends presented in the article need to be verified (see full commentary below), but the data presented in the article are nonetheless very interesting.

We thank the reviewer for the nice words and constructive comments. We fully agree that our focus is a delicate subject, and the SSC changes are the result of many influencing factors.

In response to the reviewer's suggestions, we have improved the SSC remote sensing inversion algorithm and updated the results in this revision accordingly. To ensure the accuracy of the satellite-derived SSC, we conducted comprehensive accuracy validations, including comparisons of our SSC with in-situ measured SSC, OLCI SSC products, and SSC from existing local studies. Furthermore, we utilized the Mann-Kendall test for trend analysis to ensure the reliability of the SSC trend. Please refer to the detailed response in sections 1.1, 1.2, 2.3, and 2.5.

1.1

However, the article has one major flaw: the analysis is incomplete, it does not delve deeply enough into the processes and mechanisms involved, and the comments on the results refer to and rely too much on presuppositions and general assessments. SSC and RPA have only secondary relationships and no reason to co-evolve, especially between sites.

RPA is clearly conditioned by river discharge, which is itself partly constrained by the water regulation of dams, and is very sensitive to wind. SSC depends on the river water flow, the sediment supply (Q_{river}) conditioned by the erosion of the catchment, its size and slope profile (several parameters that are integrated into the Sediment Delivery Ratio), rainfall, climate and management (dams, dredging and sand mining in particular, but also land use change). A small sloping watershed on a high island in Oceania contributes much more sediment to the sea than a larger watershed with settling and storage plains, for the same flow rate. SSC is also influenced at the coast by estuarine resuspension, stratification (which results from the ratio between buoyancy and the mixing energy provided by the tide) and resuspension by waves. The upstream estuarine transport of sediment resulting from tidal pumping, which is particularly noticeable in the dry season, brings particles from the sea into the estuary, whose fate is partially mixed with the particles brought in by the rivers. Another important factor is that the impact of dams takes time to become established before a river settles into a new hydro-sedimentary regime. While catchment areas that have had dams for decades,

such as those in the United States and Europe, have had no impact over the last twenty years because they are much older, recently installed dams in Asian catchment areas have had a major impact in reducing inputs over the last twenty years.

R1.1

Thanks for raising these detailed processes controlling RPA and SSC. We agree with the reviewer's suggestion to include additional analysis and discussion on the processes and mechanisms influencing SSC and RPA. In this study, we utilized satellite images to detect the shape of RPA and SSC on the water surface, a methodology commonly used in previous research, such as Shi and Wang (2010) and Liang et al. (2020). Many of the factors that mentioned by the reviewer are included in our analysis. Q_{river} (the sediment flux, in kg/s) integrates the effects of flow rate and sediment concentration, thereby accounting for differences between catchments. The interaction between estuarine processes and fluvial seasonality is partially included in Q_{tide} , which captures the monthly variability in tidal discharge. We also considered the response time of coastal SSC to dam construction by comparing Q_{river} data obtained from WBMSed and Dethier et al. (2022). Q_{river} from WBMSed contrasts pre-dam and modern conditions. Whereas modern time series (2000-2020) Q_{river} collected from Dethier et al. (2022), which was generated based on Landsat images, were used for trend analysis in this study. We found a significant positive correlation between WBMSed Q_{river} and Landsat Q_{river} (Fig. R1). In addition, we observed a strong relationship between the coastal SSC trends and the Landsat Q_{river} trends (Fig. 3b in main text), which might suggest a rapid response of coastal SSC to dams, yet we leave a discussion on this for future studies to keep our message concise.

Fig. R1 | Comparison of long-term mean Q_{river} derived from Landsat and WBMSed for 186 deltas examined in this study. The Landsat Q_{river} is obtained from Dethier et al. (2022). The correlation coefficients (r) and P values are annotated.

In this revision, we separately investigated the relationships between SSC and RPA with four driving factors (Q_{river} , Q_{tide} , Q_{wave} , and salinity). We found significant relationships between SSC and Q_{river} , Q_{tide} , Q_{wave} , and salinity (Fig. R2). Notably, Q_{river} also exhibited a significant correlation with RPA (Fig. R3a), as supported in other studies (Horner-Devine et al. 2015; Pritchard and Huntley 2006). However, we did not observe a significant correlation between RPA and Q_{tide} (Fig. R3b), despite the documented impacts of tides on RPA (Horner-Devine et al. 2015). This

discrepancy may arise from strong three-dimensional effects in estuaries, as also pointed out by the reviewer. For instance, tidal pumping and density-driven estuarine circulation could result in high sediment concentrations near the bed without generating a surface expression of tides on RPA (Becherer et al. 2016; Geyer and MacCready 2014; Sommerfield and Wong 2011; Van Maren et al. 2023). In addition, wave may also confine river plumes nearshore (Fig. R3c) (Flores et al. 2022). As such, high sediment concentrations do not necessarily imply larger RPAs due to the intricate movements of sediment under the integrated effects of river flow and wave and tide forces (Fig. R4) (Doxaran et al. 2009a; Mitchell 2013). Other factors that we did not explore in this study, such as the strength and direction of wind (García Berdeal et al. 2002), could also modulate the horizontal extent of river plumes and obscure the relation between SSC magnitude and RPA.

Fig. R2 | Relationships between long-term mean SSC, Q_{river} , Q_{tide} , Q_{wave} , and salinity. (a-d) The relationships between SSC and Q_{river} (a), Q_{tide} (b), Q_{wave} (c), and salinity (d). The correlation coefficients (r) and P values are annotated.

Fig. R3 | Relationships between long-term mean RPA, Q_{river} , Q_{tide} , Q_{wave} , and salinity. (a-d). The relationships between RPA and Q_{river} (a), Q_{tide} (b), Q_{wave} (c), and salinity (d). The correlation coefficients (r) and P values are annotated.

Fig. R4 | Comparisons of variations of monthly RPA and mean SSC in Yangtze and Parana at different periods with different hydrodynamics. The time, RPA, mean SSC, along with monthly maximum tidal range (MTR, unit: in m), maximum wave height (MWH, unit: in m), mean salinity (unit: in %), and mean water discharge (WD, unit: m^3/s) was tagged.

The reviewer is fully correct that not all factors are considered. Stratification for example is difficult to predict without (good) bathymetric data. We now better explain which factors we account for, and which we do not. In our discussion section, we clarify the limitation of satellite observation and emphasize the absence of a necessary correlation between RPA and SSC. In addition, we look into the dynamics of RPA and SSC by introducing processes and mechanisms that could influence their changes, supported by additional analysis. For detailed information, please refer to discussion section (Lines 182-191 and Lines 330-342). Figs. R (2-4) have all been included in this revised version to support our conclusions.

Lines 182-191:

These trend disparities are mainly due to the complexity of the spatiotemporal response of sediment plumes to SSC variations across different deltas (Extended Data Fig. 2(e-h)). For example, the SSCs in both Parana (in Brazil) under the influences of tide and Godavari (in India) decreased during 2000-2020, likely wave forces. In fact, high sediment concentrations do not necessarily imply larger RPAs due to upstream dam constructions^{22,23}, however, during the same period, their RPAs increased (Fig. 2, Extended Data Fig. 2&4). These inconsistent trends between SSC and RPA might be related to the intricate movements of sediment plumes under the integrated effects of river flow, salinity, and hydrodynamic tide and wave forces (Supplementary Figure 3(m-p))^{24,25,24,25}.

Lines 330-342:

We did not observe covariation between SSC and RPA, possibly due to limitations in satellite observations of water surface SSC and RPA. In many coastal environment, especially those governed by estuarine and wave dynamics, surface SSC can be highly localized³⁶⁻³⁸. In our study, we found significant influences of Q_{river} , Q_{tide} , Q_{wave} , and salinity on SSC dynamics (Fig. 4). Notably, Q_{river} also plays a substantial role in regulating RPA changes (Supplementary Figure 10a), as evidenced in other studies^{39,40}. However, we did not observe a significant correlation between RPA and Q_{tide} (Supplementary Figure 10b), despite the documented impacts of tides on RPA³⁹. This discrepancy may arise from strong 3-dimensional effects in estuaries. For instance, tidal pumping and density-driven estuarine circulation could result in high sediment concentrations near the bed without generating a surface expression of tides on RPA⁴¹⁻⁴⁴. In addition, wave may also confine river plumes nearshore (Supplementary Figure 10c)⁴⁵. Other factors that we did not explore, such as the strength and direction of wind⁴⁶, could also modulate the horizontal extent of river plumes and obscure the relation between SSC magnitude and RPA.

1.2

When the article mentions "the effect of sediment supply on SSC more complicated than previously assumed" (L. 24-25) or mentions "inconsistent trends" (L. 25, L. 145), the observation seems to be based on the authors' impression. To help readers find their way around, this is the perfect opportunity to list the main factors that may explain why SSC and RPA are changing either in the same direction (both increasing or both decreasing) or in opposite directions. For a given plume, if the wind comes from the sea and sticks the plume to the coast, RPA will be smaller and SSC larger than with an onshore wind. Opposite or inverse trends are not inconsistent ones.

R1.2

Thanks for your suggestions. In this revision, we are more careful in our descriptions and have corrected following reviewer's suggestion (see changes below). Furthermore, we have enhanced our analysis by examining potential factors influencing the SSC and RPA in deltas. This includes the correlation analyses between SSC, RPA, and the four key driving factors that we assessed (Q_{river} , Q_{tide} , Q_{wave} , and salinity) (Figs. R2 & 3). Furthermore, we have provided insight into the complex response of RPA to SSC by investigating variations in RPA and SSC across different tidal ranges, wave heights, salinities, and river discharges in various deltas, such as the Yangtze and Parana (Fig. R4). We have updated the main text to show these findings. Clarifications about relationships between SSC and RPA please refer to the Section R1.1.

Lines 23-27:

~~The effect of relationship between river sediment supply on and coastal SSCs may be more complicated than previously assumed, varies between deltas, with as 34.6% much as 45.2% of the deltas exhibited inconsistent showing opposing trends between river sediment supply and SSC, coastal SSCs.~~

Lines 188-191:

~~In fact, high sediment concentrations do not necessarily imply larger RPAs due to upstream dam constructions^{22,23}, however, during the same period, their RPAs increased (Fig. 2, Extended Data Fig. 2&4). These inconsistent trends between SSC and RPA might be related to the intricate movements of sediment plumes under the integrated effects of river flow, salinity, and hydrodynamic tide and wave forces (Supplementary Figure 3(m-p))^{24,25,24,25}.~~

Lines 204-205:

~~In contrast, we observed opposite trends between Q_{river} and SSC in 45.2% of the deltas (Fig. 3b).~~

Lines 210-211:

~~Furthermore, anthropogenic activities may also contribute to opposite changes between Q_{river} and SSC.~~

Lines 280-281:

~~In contrast, we observed inconsistent trends between Q_{river} and SSC in 34.6% of the deltas (Fig. 4b).~~

Lines 286-287:

~~Furthermore, anthropogenic activities may also contribute to inconsistent changes between Q_{river} and SSC.~~

1.3

In addition, the article probably contains an error in figure 2b (also Extended Data Fig. 3 & 4) that must be corrected for a later version: the units cannot be mg/L on the left, with positive or negative values (what is a SSC value < 0?), or areas on the right in km², positive or negative. The figure caption indicates "trends between 2000 and 2020": are the units mg L⁻¹ year⁻¹ or mg L⁻¹ decade⁻¹ or ...? Same concern for km² year⁻¹ or decade⁻¹. So the variables are not SSC and RPA but [SSC/Time] and [area/Time] or equivalent. This needs to be corrected. Then I wonder about the values given in the text for the resulting slopes. If the units of the parameters presented are already a quantity per unit of time (e.g. SSC/T), calculating the trend amounts is thus calculating a second derivative? This point really needs to be clarified.

R1.3

Thanks for raising this. In our previous version, the Y-axes in Figure 2b and Extended Data Figs. 3&4 represent the anomaly data of SSC and RPA. Anomaly is estimated as the differences between the monthly mean SSC or RPA and their long-term average for that month. The purpose of calculating the anomaly data is to eliminate seasonal interference in long-time series data, thereby achieving precise monitoring of long-term changes (Cai et al. 2016). Therefore, the unit should be mg/L (for SSC) or km² (for RPA), respectively. Negative or positive SSC and RPA values in the

figures indicate that the monthly mean SSC and monthly total RPA are lower or higher than their long-term average values.

In this revision, to improve the readability of the figures, we replaced the monthly anomaly with the monthly mean in Figure 2b and Supplementary Figs. (5&7). However, the long-term trends in SSC and RPA were still calculated using monthly anomalies (details are provided in the Method). We also add more details in the figure captions.

To quantify their long-term trends, we conducted trend analysis using the Mann-Kendall test in MATLAB 2021 for all SSC and RPA monthly anomaly data. We employed Sen's slope (obtained from the Mann-Kendall test) of the long-term monthly mean anomaly, multiplied by 12 (representing 12 months a year), as the mean yearly change rate to evaluate the SSC and RPA variations for each delta. The associated P value (p) was used to determine whether the changing trends were statistically significant ($p < 0.05$). Detailed description of this approach has been included in the Method section (Lines 814-829).

Lines 154-159:

Variations in monthly mean SSC and RPA, along with their long-term change trends at global scales. The orange and blue dash lines represent the long-term trends indicate variations of SSC and RPA, respectively. These trends were derived based on long-term monthly anomaly SSC and RPA, which were estimated as the difference between the monthly mean SSC and RPA and their long-term average for that month (see Method). The Sen's slope and P p -value were obtained from the Mann-Kendall test are annotated.

Lines 814-829:

Trend analysis. The SSC and RPA monthly mean anomaly, ~~estimated~~ determined as the differences between the monthly mean SSC, RPA, and Q_{river} and RPA and their corresponding long-term average for that specific month, ~~was~~ were used to examine the long-term change trend in these ~~three~~ two parameters over the past two decades. ~~We adopted~~ This anomaly processing is primarily used to remove the seasonal interference in long-time series data, thereby achieving precise monitoring of long-term changes⁸⁷. We employed Sen's slope (obtained from the Mann-Kendall test) of the long-term monthly ~~anomaly~~ mean anomaly, multiplied by 12 (representing 12 months a year), as the mean yearly change rate to evaluate the SSC, RPA and Q_{river} RPA variations for each delta, and ~~used~~ the associated P value (p) (~~with t test~~) was used to assess whether the changing trends were statistically significant ($p < 0.05$). The monthly anomaly SSC and RPA at global or continental scales were generated by using the average monthly SSC and RPA from 349 deltas or from deltas located on different continents.

Furthermore, the Sen's slope (Mann-Kendall test) of the annual mean Q_{river} , Q_{tide} , Q_{wave} , and Salinity was adopted to indicate the long-term changes in Q_{river} , Q_{tide} , Q_{wave} , and Salinity, with associated p-value used to assess the significance of the change trends. The trend analysis using Mann-Kendall was conducted in MATLAB 2021.

1.4

The section comparing SSC and the "river sediment supply" (L. 186) clearly illustrates that SSC is far from representing the sediment supply. Global values are given without comment. The discussion would benefit from an introduction to the 3D processes occurring at river mouths. For example: the sediment supply integrates inputs from the surface to the bottom and the three-dimensional effects in the estuary and in the plumes are major. While there is an export to the sea at the surface, there is often an import of sediment to the land in the salt wedge generated by tidal pumping, when there is any.

R1.4

We fully agree with the reviewer that there are many three-dimensional processes in estuaries that complicate a straightforward relation between sediment supply and SSC. These processes cannot all be assessed in this global study because a complete global forcing and validation data do not exist. However, to address this issue, we now further explain these processes and how they affect our results, by including additional discussions with studies from Eisma and Kalf (1984), Bates (Bates 1953), and others (Becherer et al. 2016).

One important remark we would like to make is that we look at temporal trends in SSC and their underlying drivers, such as river sediment supply changes. Despite all the complications from three-dimensional processes, one would still expect that an increase in river sediment supply would lead to an increase in surface SSC, assuming all other factors remaining constant. We observed that at certain individual deltas, coastal SSC are closely related to Q_{river} variations, as has also been shown elsewhere (Cao et al. 2022; Luo et al. 2022). However, not all deltas exhibited consistent changes between SSC and Q_{river} (Fig. R5b), primarily due to the trends arising from other processes (e.g., tides, waves, salinity, and human interventions such as dredging) (Edmonds et al. 2023; Kemp et al. 2020; Syvitski et al. 2005; Van Maren et al. 2016). This is confirmed in other studies. For instance, although Q_{river} has declined due to upstream dam construction in the Mekong, seasonal coastal SSC in the region has increased due to wave-driven suspended sediment (Loisel et al. 2014). Moreover, tide-induced suspended sediment transport is often achieved through tidal pumping, attributed to current asymmetry and sediment lag effects (Brenon and Le Hir 1999; Dyer 1973; Uncles and Stephens 1989). This process can be seen as a competition between downstream river transport and upstream transport induced by tidal asymmetry (Yu et al. 2014). While there is a surface export of river sediment to the sea, tidal pumping frequently results in sediment influx towards the land within the salt wedge, causing elevated turbidity landward of the estuary (Becherer et al. 2016; Sommerfield and Wong 2011; Van Maren et al. 2023). This phenomenon is especially prominent in well-mixed and macro-tidal estuaries, where tidal pumping governs sediment transport under calm conditions (Becherer et al. 2016; Yu et al. 2014). Even in deltas with hyperpycnal plumes, the SSC at the water surface is anticipated to be low. However, due to the influence of tidal currents, sediment retention and SSC amplification may occur (He et al. 2022; Shanmugam 2018).

Fig. R5 | Change trends of Q_{river} and SSC between 2000 and 2020 for 186 deltas. (a). Spatial patterns of the trends (increase or decrease) in Q_{river} and SSC; (b). Scatterplot of change rates (Sen's slopes from *Mann-Kendall test*) in Q_{river} and SSC for 186 deltas. Each quadrant (Q1, Q2, Q3, and Q4) represents a combination of changing trends between SSC and Q_{river} . The numbers in parentheses represent the percentages of deltas.

We have now provided detailed clarification in the main txt (Lines 195-203 and Lines 343-357).

Lines 195-203:

We observed consistent variations between Q_{river} and SSC in 54.8% of examined deltas (Fig. 3). Specifically, among these 54.8% of deltas, 32.8% are gaining SSC as well as Q_{river} . Many of these deltas are larger river- and tide-dominated deltas, such as the Amazon and the Mississippi (Fig. 3b). Consequently, these consistent increments may be attributed to the large river inputs and tidal forces, which then outweigh other controls on SSC and make the effects readily observed (Fig. 3(c-d))(see also in Ref. ²⁶ and Ref. ²⁷). Meanwhile, 22% of the deltas display reductions in both SSC and Q_{river} , primarily located in South Asia and the southeast of South America (Fig. 3(a-b)). These deltas exhibit lower Q_{river} and higher Q_{wave} and salinity (Fig. 3(c-d)).

In contrast, we observed opposite trends between Q_{river} and SSC in 45.2% of the deltas (Fig. 3b). Among these, 30.1% of the deltas exhibited an increase in SSC while Q_{river} declined, and 15.1% of the deltas showed a decline in SSC while Q_{river} increased. Many of these deltas had lower Q_{river} and weak tidal forces, or higher salinity and strong wave energy (Fig.3(c-f)). These findings highlight the complexity of coastal sediment in response to Q_{river} . Other controls, such as wind, wave, or tidal currents, as well as salinity might obscure the response of SSC to Q_{river} ^{16,28,29}. Furthermore, anthropogenic activities may also contribute to opposite changes between Q_{river} and SSC. For example, channel deepening and dredging activities have resulted in high SSC in the Ems delta, despite its small and decreasing Q_{river} ³⁰.

Lines 321-329:

~~We observed that at certain individual deltas, coastal SSC are closely related to Q_{river} variations, as has also been shown elsewhere ^{20,36}. However, not all deltas exhibited consistent changes between SSC and Q_{river} , primarily due to the complex influences from hydrodynamic forces (i.e., tide and wave), delta morphology, and human interventions (i.e., sediment dredging) ^{9,37-39}. For instance, although Q_{river} has declined due to upstream dam construction in the Mekong, seasonal coastal SSC in the region has increased due to wave driven suspended sediment ¹⁹. Our findings highlight an intricate relationship between coastal SSC and Q_{river} .~~

Lines 343-357:

We observed that near certain deltas, coastal SSC are closely related to Q_{river} variations, as has also been shown elsewhere^{21,47}. However, not all deltas exhibited consistent changes between SSC and Q_{river} (Fig. 3(a-b)), likely due to the complex influences from hydrodynamic forces (e.g., tide and wave), salinity, and human interventions (e.g., sediment dredging) (e.g., sediment dredging)^{9,48-50}. For instance, although Q_{river} has declined due to dam construction in the Mekong, there has been wave-driven enhancement of seasonal coastal SSC²⁰. Additionally, river plume dynamics induced by river flow density may also complicate the relationship between Q_{river} and SSC. Increases in Q_{river} might densify the river flow, stimulating the development of hyperpycnal plumes without generating a surface expression in the SSC⁵¹⁻⁵³. For example, in the Huanghe delta, nearly 80% of the river sediment on the delta front is deposited from hyperpycnal flows⁵⁴, making surface SSC signature hard to express. However, satellite imagery is easier to capture the distinct surface SSC signature for low-density hypopycnal plumes^{51,55,56}. ~~Our detailed analysis of dynamics of coastal SSC and its response to Q_{river} variations can provide valuable insight for predicting the future of.~~ As such, the satellite observed variations in SSC from the water surface are therefore difficult to correlate with changes in river sediment supply (Fig. 3b).

1.5

It should also be noted that the Q_{river} values used come from an article submitted but not yet published (Sun et al., Nature climate change). The methodology raises questions: how did these authors include vertical profiles (stratified or not) in their estimate of Q_{river} based solely on surface suspended sediment data from Landsat data (L. 454)? Did you try to compare their SSC from Landsat with yours from MODIS? Until this article is published, it is impossible to assess the methodology used in the present version. The same applies to Q_{tide} , whose calculation formula uses Q_{river} collected from Sun et al (submitted).

R1.5

Thanks for raising this. In this revised version, we have switched to using the annual (from 2000-2020) river sediment flux (Q_{river}) from Dethier et al. (2022), which was published online (<https://zenodo.org/records/7808492>). This sediment flux is generated by integrating the river SSC observed from Landsat (from 1984 to 2020) with the water discharge data obtained from corresponding hydrological stations along the rivers. The underlying assumption is that the SSC throughout the water column remains consistent with the surface. While we recognize the importance of accounting for variations in SSC across vertical profiles, the estimation was limited by the surface observation capabilities of satellite images. Presently, the predominant method for satellite-based sediment flux relies on this approach.

Among the 349 examined deltas of our study, 186 deltas were successfully matched with the Q_{river} dataset provided by Dethier et al. (2022). We now provide the correlations between Q_{river} and coastal SSC using this data (see new Figure 3) and no longer refer to the publication of Sun et al.. We have revised the descriptions regarding the collection of Q_{river} and water discharge data, please refer to Lines 594-599 and Lines 606-607.

Additionally, as a test, we conducted a comparison of the long-term mean coastal SSC (MODIS, of this study) and river SSC (from Landsat, Dethier et al. (2022)) for 186 rivers, along with their long-term change trends (Fig. R6). But, we did not find a statistically significant trend (Fig. R6a).

Furthermore, the change trends observed in SSC derived from MODIS did not entirely align with those derived from Landsat (Fig. R6b). These discrepancies are likely because of the influence of the river size/flow rate. A high SSC in a small river creates a small coastal SSC. We therefore decided not to include this in the paper to keep our message concise.

Fig. R6 | Comparison of the long-term mean SSC derived from MODIS and Landsat, along with their respective change trends, for 186 deltas. (a). Comparison of long-term mean SSC derived from MODIS and Landsat, the correlation coefficients (r) and P values were annotated; **(b).** Scatterplot of change rates (Sen's slopes from *Mann-Kendall test*) in SSC from MODIS and Landsat for 186 deltas. Each quadrant (Q1, Q2, Q3, and Q4) represents a combination of changing trends between SSC from MODIS and Landsat. The numbers in parentheses represent the percentages of deltas in the specific quadrant relative to the total examined deltas.

Lines 594-599:

River, tidal, and wave sediment flux. We obtained the ~~monthly~~ annual river sediment flux (Q_{river}) between 2000 and ~~2018~~2020 from a global fluvial sediment flux dataset created by Ref. 66. This dataset was generated using the ~~monthly mean~~ river SSC derived from Landsat images and water discharge from matched river discharge stations. The water discharge (measured in cubic meters per second, m³/s) represents the fluvial flow passing through the entire river cross-section. Out of the 349 examined deltas, ~~191~~186 deltas were successfully matched with Ref. 66's dataset.

Lines 606-607:

Where Q_{river} is fluvial sediment flux (kg/s) and Q_{river} ~~is~~ denotes water discharge (m³/s), both of which are ~~all collected~~ obtained from Ref. 66.

1.6

A separate analysis of trends by delta morphology type could be instructive. The fine fraction dominates at the surface in SSCs, and is more likely to be deposited (and resuspended) in tide-dominated deltas (which are generally dominated by fine sediments) than in wave-dominated deltas (where coarse sediments dominate, and where fine sediments are therefore not deposited because of excess energy). It should be noted, however, that as a result of changes in river inputs, it is quite possible that some river-dominated deltas may have recently become wave- or tide-dominated, due to the strong decrease in river sediment supply.

R1.6

Thanks for the suggestions. In addition to the previously provided SSC distribution among three delta morphologies, in this revision, we have incorporated additional analyses. These include the SSC and RPA distribution and their change trend analyses among different delta morphologies (Fig. 1 in main text, Fig. R7). Moreover, when examining the relationship between long-term mean SSC and Q_{river} , Q_{tide} , Q_{wave} , and salinity, we analyzed the correlation relationships among delta morphology (Fig. R2). Similarly, we differentiated morphology types when analyzing the response of SSC to Q_{river} (Fig. R5).

Conducting in-depth analyses based on delta morphology has allowed us to gain a clearer understanding of the significant differences in SSC distribution among different morphologies, thereby enhancing the persuasiveness of our results. We have also elucidated the variations in SSC and RPA distribution across different delta morphologies in several sections of the main txt. Please refer to Lines 95-98, 119-122, 146-148, 173-175, 196-198, and 237-248.

Fig. R7 | Comparison of the change rates of SSC and RPA between the Arctic (>50°N) and non-Arctic river deltas, as well as different delta morphologies. (a-b). Density histograms of the change rates (Sen's slope from *Mann-Kendall test*) of SSC (a) and RPA (b) in Arctic and non-Arctic river deltas; **(c-d).** Density histograms of the

change rates of SSC (a) and RPA (b) among different delta morphologies. The bar chart in the panel illustrates the percentage of deltas presenting different change trends (significant increase or decrease, increase or decrease). The mean change rates for SSC and RPA are indicated in parentheses following the legend. All plots were generated using Origin 2021.

Lines 95-98:

Additionally, the long-term mean SSC also varied among different delta morphologies (Fig. 1c), with the highest SSC observed in tide-dominated deltas (median SSC of 4235.1 mg/L and 55.0 mg/L, respectively), while the lowest was found in wave-dominated deltas (median of 16.3 mg/L).

Lines 119-122:

Heterogeneities were also observed across different delta morphologies, as both the river-dominated (median of RPA: 1182.155.8 km²) and Delaware (mean of SSC: 95.9 mg/L; mean tide-dominated (median of 52.3 km²) deltas exhibited high RPA, while the lowest RPA was found in wave-dominated deltas (median of RPA: 857.0 km²), highlighting the spatial heterogeneities between local coastal SSC and sediment plumes. 37.7 km²) (Supplementary Figure 4b).

Lines 146-148:

Among three delta morphologies, the most pronounced increase was found in river-dominated deltas, with 62.0% of these deltas displaying an increasing trend and having a mean rate of +0.36 mg/L yr⁻¹ (Supplementary Figure 6c).

Lines 173-175:

Meanwhile, over 62% of both river-dominated and tide-dominated deltas exhibited an increasing trend in RPA, with a mean rate of over +0.8 km² yr⁻¹, more than double that of wave-dominated deltas (Supplementary Figure 6d).

Lines 196-198:

Specifically, among these 54.8% of deltas, 32.8% are gaining SSC as well as Q_{river} . Many of these deltas are larger river- and tide-dominated deltas, such as the Amazon and the Mississippi (Fig. 3b).

Lines 237-248:

This is evident from the fact that wave-dominated deltas showed the lowest median SSC (39.5 mg/L) compared to river-dominated (median SSC of 85.0 mg/L) and tide-dominated (median SSC of 101.3 mg/L) deltas (Extended Data Fig. 61). Moreover, the wave-dominated deltas are more likely to have a decreasing SSC compared to river- and tide-dominated deltas (Supplementary Figure 6c). This discrepancy between seasonal intra-delta and long-term inter-delta SSC dynamics suggests that wave-dominated deltas, characterized by the absence of distributary networks and sandy shorelines³¹²⁶ (Extended Data Fig. 6 (Supplementary Figure 2(a-f))), may do not possess efficiently retain coastal sediment retention abilities. Waves increase SSCs temporarily, but the long-term effect of higher wave- dominance is a decrease in SSC. In tide- and river-dominated deltas, on the other hand, seasonal and long-term SSCs change in unison the same pattern, showing their capacity to retain coastal sediments nearshore for extended periods.

1.7

The discussion introduces generalities that are either unnecessary or too vague:

R1.7

Thank you for your suggestion. We have made point-by-point modifications following your suggestions below.

1.8

- L.230-231: "Our analysis... can provide valuable insight for predicting the future of coastal wetlands"). Which kind of prediction can be done using the article?

R1.8

We have clarified this as "The global SSC dynamics that we observe and their dependence on Q_{river} offers valuable insights into sediment availability for coastal wetlands. Coastal SSC is a vital predictor when evaluating the resilience of coastal wetlands in the face of sea-level rise (Schuerch et al. 2018). Previous research has raised concerns that the decline in Q_{river} could potentially limit the availability of coastal SSC, thereby threatening the survival of wetlands (Lovelock et al. 2015). To protect coastal wetlands, various studies have therefore proposed removing dams to enhance Q_{river} and subsequently increase coastal SSC (Bednarek 2001; Xie et al. 2020). However, this may be less straightforward than previously thought. Our results indicate that not all Q_{river} declines lead to SSC loss, at least within the timeframe of our analysis (Fig. 4b). In addition, we observed a prominent increase in coastal SSCs over the past two decades. This could enhance wetland resilience to future sea-level rise, and wetland vulnerability might be overestimated from previous studies (Kirwan et al. 2016).

Lines 358-370:

The global SSC dynamics that we observe and their dependence on Q_{river} offers valuable insights into sediment availability for coastal wetlands. Coastal SSC is a vital predictor when evaluating the resilience of coastal wetlands in the face of sea-level rise ^{19,48}. ~~The common concern is.~~ Previous research has raised concerns that Q_{river} the decline ~~would likely constrain coastal SSC in~~ Q_{river} could potentially limit the availability, ~~ultimately~~ of coastal SSC, thereby threatening ~~wetlands~~ the survival of wetlands ²². To protect coastal ~~wetland~~ wetlands, various studies have therefore proposed ~~to remove~~ removing dams to enhance Q_{river} and subsequently increase coastal SSC ^{4,57,40}. However, this may be less straightforward ~~that~~ than previously thought. Our results indicate that not all Q_{river} declines will lead to ~~coastal sediment deficits~~ SSC loss, at least within the timeframe of our analysis (Fig. 4b). In addition, we observed a prominent increase in coastal SSCs over the past two decades. ~~These might offer a hopeful perspective on coastal Future wetland preservation.~~ This could enhance wetland resilience to future sea-level rise, and wetland vulnerability might be overestimated from previous studies (Kirwan et al. 2016) ~~preservation.~~

1.9

- L. 233-234, "the common concern is that Q_{river} decline would likely constrain SSC availability": please refer to the poor correlation between Q_{river} and SSC (Fig. 3a).

R1.9

Here, we aim to convey that previous study suggests a decline in Q_{river} could limit the availability of SSC in estuaries. We have revised the expression "Previous research has raised concerns that the decline in Q_{river} could potentially limit the availability of coastal SSC, thereby threatening the survival of wetlands (Lovelock et al. 2015).".

Lines 360-363:

Previous research has raised concerns that Q_{river} the decline ~~would likely constrain coastal SSC~~ in Q_{river} could potentially limit the availability, ~~ultimately~~ of coastal SSC, thereby threatening ~~wetlands~~ the survival of wetlands ²².

1.10

- L. 270-273 "our results ... can aid in forecasting the fate of ...". How so?

R1.10

We have clarified this as "Our results offer valuable insights into the present distribution and dynamics of coastal sediment near river deltas. The SSC dataset can be utilized to assess sediment balance around coastal wetlands, thereby objectively evaluating the threat of sea-level rise. Furthermore, it could offer crucial information for the protection and restoration of coastal areas, especially those experiencing sediment deficits".

Lines 398-404:

Our results offer valuable ~~baseline information for understanding~~ insights into the ~~current present~~ distribution and dynamics of coastal ~~sediments. This~~ sediment near river deltas. The SSC dataset can ~~aid in forecasting the fate of~~ be utilized to assess sediment balance around coastal wetlands ~~threatened by,~~ thereby objectively evaluating the threat of sea-level rise ~~and support policy decisions regarding.~~ Furthermore, it could offer crucial information for the protection and restoration ~~and protection~~ of coastal ~~ecosystems~~.

1.11

These general statements are inaccurate, unsubstantiated, unnecessary, and confusing. The results of the study, if well presented, will suffice to support a very informative article from a scientific point of view and on the dynamics underway in front of river mouths without leading to such unsupported generalities.

R1.11

Thanks for your suggestion. We have modified these statements following your suggestions.

1.12

Overall, the article contains some very interesting information, which needs to be corrected (Fig. 2b). The explanations and comments are not explicit enough, ignore too much the diversity of catchment areas and 3D processes at river mouths, and are sometimes based on unsubstantiated assumptions. First of all, replace "inconsistent trends" with "opposing trends", and you will be in a more suitable position to detail the hidden mechanisms and take account of their wide diversity.

R1.12

Thanks for your suggestion. We have revised Figure 2 and conducted an in-depth analysis. Additional figures and tables have been incorporated for clarification, aiming to enhance the persuasiveness of our conclusions. Following your suggestion, we have revised all expressions of "inconsistent trends" in the article, see R1.2.

We agree with your opinion that greater discussion on relevant three-dimensional processes can help to better explain our findings. In this revision, we primarily conducted in-depth analysis and discussion based on our research findings and literature review (see R1.1 & 1.4). We hope that in the future, with more additional data becoming available, we can incorporate large-scale three-dimensional simulations to facilitate more comprehensive analyses.

1.13

This article cannot be published as it stands, but it deserves some in-depth work before publication.

R1.13

Thanks for your kind comments. We sincerely hope our revisions have addressed all of your concerns.

Reviewer #2 (Remarks to the Author):

Summary and overall appreciation

This study uses 20 years of satellite data (MODIS surface reflectance, monthly composites at 500 m spatial resolution, from 2000 to 2020) to reconstitute the evolution of suspended sediment concentration (SSC) and river sediment plume area (RPA) over 349 deltas worldwide. Results are reported globally then distinguishing the 6 continents. Temporal trends are computed and related to the respective influence of tides, waves and delta morphologies. The results obtained are rather unexpected: an increase of coastal SSC is revealed despite a recognized “decline in global river sediment supply to the coast over the past few decades.”

The study is of high interest: coastal environments, notably those influenced by river inputs, are highly impacted by climate change effects and human activities and must be carefully monitored. Other studies (e.g., cited literature and references therein) already addressed this topic notably using ocean color satellite data, but usually at regional scales or globally over a shorter time period. Results and analyses in the present studies provide some new insights but also rise questions. The amount of work (e.g., number of satellite images processed and analyzed) is impressive, however several methodologies issues should be clarified to fully appreciate the results and conclusions of the study.

We very much appreciate the reviewer’s positive evaluation of our study. Thank you for your comments.

2.1

Detailed questions and comments are provided hereafter to improve the study and current version of the manuscript.

‘.decline in global river sediment supply to the coast over the past few decades’

This is not obvious and strongly regional (Li et al. 2020) e.g., with the north-south divergence and shift from Asia to South America stated by Dethier et al. (2022)

R2.1

Thanks for your comment. We have modified this expression as “Therefore, concerns have been raised about wetlands’ fate due to the decline of sediment supply to the coast in many global rivers”.

Lines 14-15:

Therefore, concerns have been raised about ~~their~~ wetlands’ fate due to ~~increased recognition of~~ the decline ~~in global~~ of river sediment supply to many deltas.

Lines 33-35:

In recent decades, various global-scale assessments have highlighted a decline in ~~river~~ sediment supply ~~from many rivers~~ due to the construction of river dams ⁶⁻¹⁰⁶⁻¹⁰.

2.2

Main, 1st paragraph: what about coastal erosion??

R2.2

Thanks for raising this. Coastal erosion can indeed be a contributor to short-term coastal SSC variation (Russell 1993). We have now further clarified this in the main text. Please refer to Lines 39-42.

Lines 39-42:

A possible explanation is that the coastal SSCs are also affected by physical feedback between coastal hydrodynamics (e.g., river flow, tides, and waves) and sediment transports (e.g., suspension, erosion, deposition, and movement) ¹⁴⁻¹⁷~~14-16~~.

2.3

Methods:

Why not using the Han et al. (2016) or Wei et al. (2021) global SPM algorithm?

Han B, et al. Development of a Semi-Analytical Algorithm for the Retrieval of Suspended Particulate Matter from Remote Sensing over Clear to Very Turbid Waters. Remote Sensing. 2016; 8(3):211. <https://doi.org/10.3390/rs8030211>

More importantly would your results be similar or different if using their SPM algorithm? In other words, are your results (temporal trends) dependent of the SPM estimation method you used? This is a crucial point to be clarified.

R2.3

We agree with the reviewer's suggestion. In this revision, we have now assessed the SSC inversion accuracy among algorithms from Han et al. (2016), Yu et al. (2019), and Feng et al. (2014). We did not evaluate the global coastal SSC algorithm from Wei et al. (2021), primarily because it is an improvement of the algorithm in Yu et al. (2019), which adds two additional blue bands based on the algorithm in Yu et al. (2019) to improve the SSC inversion accuracy in clear water. However, these two blue bands are not available in MODIS surface reflectance (SR) products. Additionally, we did not assess the global algorithm developed by Balasubramanian et al. (2020), mainly because this algorithm heavily depends on one of the inherent optical properties (IOP)-particulate backscattering, which cannot be directly derived from the MODIS SR data.

The reflectance data utilized in Han et al. (2016), Yu et al. (2019), and Feng et al. (2014) (whose algorithm we had previously used) were all remote sensing reflectance (R_{rs}) and were generated using an atmospheric correction approach tailored for ocean color application (Wang and Shi 2007). This R_{rs} differs from MODIS SR, which was generated using a land-based atmospheric correction algorithm. Furthermore, the algorithm in Yu et al. (2019) was primarily designed for VIIRS satellite data, meaning its spectral response function for certain bands differs from that of MODIS. Consequently, we initially used our collected in-situ measured SSC and the corresponding daily MODIS Aqua SR data to recalibrate the algorithm parameters for these three algorithms. Subsequently, we assessed the accuracy of these algorithms. The processing details are outlined below:

Han_adapted. We utilized the red (645 nm) and NIR (859 nm) bands from MODIS SR products to replace the red (671 nm) and NIR (745 nm) bands in Han et al. (2016) for SSC retrieval. In the

adapted algorithm, we employed two thresholds (0.04 and 0.09) based on the red band reflectance (R_{645}) as the blending boundaries. These two thresholds were determined based on our collected in-situ measured SSC and the corresponding red reflectance from MODIS Aqua SR. Therefore, the algorithm was defined as follows:

$$SSC_{clear} = A_T(\lambda_1)\rho_W(\lambda_1)/(1 - \rho_W(\lambda_1)/C(\lambda_1))$$

$$SSC_{turbid} = A_T(\lambda_2)\rho_W(\lambda_2)/(1 - \rho_W(\lambda_2)/C(\lambda_2))$$

$$SSC_{final} = \frac{W_c * SSC_{clear} + W_t * SSC_{turbid}}{W_c + W_t}$$

$$W_c = \log_{10} 0.04 - \log_{10}(R_{645})$$

$$W_t = \log_{10}(R_{645}) - \log_{10} 0.03$$

Where $A_T(\lambda)$ and $C(\lambda)$ are wavelength-dependent coefficients, with λ_1 and λ_2 are set to 645 and 859 nm, respectively. The values of $A_T(\lambda)$ and $C(\lambda)$ for λ_1 are 228.1 and 0.164, and for λ_2 are 3078.9 and 0.211. These coefficients are obtained from Dogliotti et al. (2015). The parameters W_c and W_t represent the weights for the SSC estimations from clear to turbid waters, respectively. These weights are determined based on the value of R_{645} . Specifically, if $R_{645} < 0.04$, W_c is 1 and W_t is 0. Conversely, if $R_{645} > 0.09$, W_c is 0 and W_t is 1. For R_{645} values between these two boundaries, W_c and W_t are calculated using the formula above.

Feng_adapted. We replaced the single band reflectance (645 nm) in Feng et al. (2014) with a band ratio (R_{645}/R_{555} , which is the ratio of SR between MODIS 645 and 555 nm bands) for sediment retrieval in relatively clear waters. Additionally, we utilized the ratio between 859 and 645 nm (R_{859}/R_{645}) for sediment retrieval in highly turbid waters. The algorithm was defined as follows:

$$SSC_{Low} = 0.82 * e^{4.62 * (R_{645}/R_{555})}$$

$$SSC_{High} = 336.23 * (R_{859}/R_{645})^2 + 387.58 * (R_{859}/R_{645})$$

$$SSC_{Middle} = \alpha * SSC_{low} + \beta * SSC_{high}$$

$$SSC(\text{mg/L}) = \begin{cases} SSC_{Low} & (SSC_{Low} < 50 \text{ mg/L}) \\ SSC_{High} & (150 \text{ mg/L} < SSC_{Low}) \\ SSC_{Middle} & (50 \text{ mg/L} \leq SSC_{Low} \leq 150 \text{ mg/L}) \end{cases}$$

Similar to Feng et al. (2014), we defined the α as $\ln(150/SSC_{Low})/\ln(150/50)$, and β as $\ln(SSC_{Low}/50)/\ln(150/50)$. In this adapted algorithm, the SSC_{Low} was first calculated using R_{645}/R_{555} , and then the retrieval algorithm was determined based on SSC_{Low} . For the relatively clear water ($SSC < 50 \text{ mg/L}$), SSC_{Low} was adopted, and for water with $SSC > 150 \text{ mg/L}$, the SSC_{High} was selected. For the intermediate SSC values, a mixture of SSC_{Low} and SSC_{High} was applied to remove the discontinuity between these two algorithms.

Yu_adapted. The blue (R_{469}), green (R_{555}), red (R_{645}), and NIR (R_{859}) bands from MODIS SR products replaced the bands at 486, 551, 671, and 862 nm utilized in Yu et al. (2019) for SSC inversion. A band of 745 nm used by Yu et al. (2019), which was manually interpolated, was not used in this study. We calibrate the parameter utilized in Yu et al. (2019) using the least squares error minimization principle, resulting in the algorithm as follows:

$$SSC = \exp(0.859 * (0.145 * (R_{555}/R_{469}) + (5.167 * (R_{645}/R_{555}) * (R_{645}/(R_{645} + R_{859}))) + (7.244 * (R_{859}/R_{555}) * (R_{859}/($$

$$R_{645+R_{859}})^{0.990}$$

Subsequently, we compared the retrieved SSC from these three adapted algorithms mentioned above with in-situ measured SSC. We found that the Yu_adapted algorithm demonstrated the highest accuracy among the three algorithms (Table R1). Consequently, the Yu_adapted algorithm was chosen for SSC inversion in this research.

Table R1 | Comparison of the accuracy between the existing SSC inversion algorithm and our algorithm. The number (n) in parentheses represents the sample size.

With		Han_adapted		Feng_adapted		Yu_adapted	
		RMSE(%)	MRE(%)	RMSE(%)	MRE(%)	RMSE(%)	MRE(%)
SSC	SSC<50 mg/L (n=496)	63.23	78.53	28.51	53.68	24.87	49.51
over	SSC>=50 mg/L (n=18)	55.81	71.92	38.65	61.11	75.53	77.32
500	Total	56.50	51.99	28.51	28.82	25.32	24.80
mg/L							
Without		Han_adapted		Feng_adapted		Yu_adapted	
		RMSE(%)	MRE(%)	RMSE(%)	MRE(%)	RMSE(%)	MRE(%)
SSC	SSC<50 mg/L (n=496)	63.23	78.53	28.51	53.68	24.87	49.51
over	500>=SSC>=50 mg/L						
500	(n=13)	68.94	60.32	48.64	64.28	49.06	53.86
mg/L	Total	56.04	51.93	28.64	29.08	24.90	24.56

However, we noted significant uncertainty in the Yu_adapted algorithm when applied to turbid water bodies (e.g., >500 mg/L), possibly due to the limited number of in-situ samples from turbid water bodies. Despite collecting a large amount of in-situ SSC data, interference from clouds and coarse spatial resolution of MODIS led to the land adjacency effect, resulting in only a relatively small number of in-situ data being matched with satellite observations. Given the long-term average coastal SSC of the majority of deltas globally does not exceed 500 mg/L (Wei et al. 2021), to mitigate the impact of high-turbidity water bodies on the algorithm, we assumed that SSC inverted by the Yu_adapted algorithm exceeding 500 mg/L would likely have considerable errors and thus should be excluded. After removing samples with concentrations exceeding 500 mg/L, we reassessed the algorithm's accuracy. The overall accuracy of the algorithm and its accuracy in high-value regions both improved (Table R1). Therefore, only the inverted SSC lower than 500 mg/L was considered reliable and used for further analysis. We chose 500 mg/L instead of 100 mg/L because while the monthly mean SSC in most deltas is likely below 100 mg/L, some deltas may indeed experience high SSC at times. Limiting SSC to 100 mg/L would therefore result in a lack of data for many of these deltas.

To further validate the accuracy of Yu_adapted, we first compared the ranges of our MODIS-derived SSC to the previously reported SSC through a review of published literature. In total, 31 deltas across the globe were examined (Supplementary Table 2). We observed that our MODIS-derived SSC aligns well with the published SSC for most deltas, yet discrepancies also exist in some deltas. Considering the disagreements of the investigation time and satellite sensor selections, these disparities are considered acceptable. Additionally, we compared the multi-year average SSC between 500-m MODIS and 4-km Sentinel-3 OLCI, noting that the average SSC from MODIS compared to OLCI has a mean ratio of 1.41 ± 0.83 (Fig. R8). However, when SSC exceeds 100 mg/L, the average SSC from MODIS is noticeably higher than that from OLCI. Given the

differences in spatial resolution, we believe such discrepancies are acceptable. Furthermore, we separately compared the temporal variations in SSC from 2016 to 2020 for several typical estuaries, ranging from clear to turbid. We observed that the trends based on MODIS and OLCI are very similar (Fig. R9). Consequently, we applied the Yu_adapted algorithm to MODIS Aqua and Terra 8-day SR products to retrieve coastal SSC for the 349 deltas.

Fig. R8 | A comparison between the multi-year (2016-2020) average MODIS-derived SSC (500 m resolution) in this study and the OLCI SSC (4 km resolution). The average ratio here represents the mean value of the ratio between the average SSC from MODIS and the average SSC from OLCI.

Fig. R9 | Comparisons of monthly mean time series (from April 2016 to December 2020) between MODIS-derived SSC in this study and the corresponding OLCI SSC among different river deltas (ranging from clear to turbid). The mean ratio in (a-d) represents the ratio of MODIS SSC and OLCI SSC.

We have added the descriptions above in the main txt (see Lines 649-752) and Supplementary Note 1 in the Supplementary material.

Lines 649-752:

To retrieve coastal SSC with a wide dynamic range (e.g., from clear to turbid) using satellite images, numerous algorithms have been developed over the past decades⁷⁵⁻⁷⁸. Nevertheless, most of these algorithms require a reflectance threshold from specific wavelengths as the blending boundaries, which varies among different research. Recently, several algorithms for estimating global coastal sediment have been proposed^{18,22,79}. ~~Over the past decades, a single red band was often utilized for near surface SSC retrievals in site specific studies¹⁰⁻¹², which usually demonstrated accurate estimates within relatively clear waters (e.g., $SSC < 100$ mg/L). Nevertheless, when water becomes extremely turbid (e.g., $SSC > 1000$ mg/L), the red band will be insensitive to SSC and even exhibit a saturation effect¹³⁻¹⁷. Under this circumstance, the red band combined with the near-infrared (NIR) band or even shortwave infrared (SWIR) band was often used for SSC retrieval by many studies¹⁸⁻²¹. For example, a soft-switching algorithm proposed by Ref.²² has successfully used both red (645 nm) and NIR (859 nm) bands from MODIS to retrieve SSC in the Yangtze River estuary, which demonstrated an uncertainty around 20-30%. Herein, we firstly~~, enabling the quantification of coastal sediment at global levels. In this study, we conducted accuracy assessments for these algorithms based on in-situ measured data and MODIS surface reflectance (SR) data to select the most accurate algorithm for SSC inversion.

Initially, we selected ~~match-ups~~ matchups between the daily MODIS Aqua SR data and in-situ SSC data ~~on~~ for the same day, ~~and excluded~~ excluding samples within three pixels ~~away from the~~ of land-water boundaries or clouds to ~~avoid~~ prevent land or cloud contaminations, ~~resulting~~. This resulted in a total of 509 matchups. Subsequently, we used these matchups to conduct a recalibration (details see Supplementary Note 1) of several typical existing coastal SSC inversion algorithms, including the algorithm from Ref.⁷⁵ (hereafter as Han_adapted), Ref.⁷⁶ (hereafter as Feng_adapted), and Ref.²² ~~505 match-ups. Then, we assessed the performance of Ref. 22's algorithm using these match-ups. However, the performance was poor when applying~~ (hereafter as Yu_adapted), and assessed their accuracy. We found that after calibrating algorithm parameters based on our in-situ measured data and MODIS SR data, Han_adapted had the largest error (overall accuracy > 51%), with RMSE exceeding 50% in both clear (<50 mg/L) and turbid water (≥ 50 mg/L) (Supplementary Table 1). Feng_adapted achieved a better overall accuracy of around 28%, with accuracies of 28.5% for clear water and 48.6% for turbid water. The model with the highest accuracy was Yu_adapted, with an overall accuracy of 24.9%, particularly performing well in clear water (RMSE=24.9%), and 49.1% for turbid water (Supplementary Figure 1 & Table 1). Considering the SSCs of the majority of deltas are not extremely high, in ~~this algorithm to global coastal regions. We attributed this failure to two aspects: 1) Ref. 22's algorithm was calibrated using limited SSC samples collected from the Yangtze River estuary, therefore might not be representative of other river mouths; 2) Ref. 22's study, we adopted the Yu_adapted~~ algorithm was developed based on MODIS Rrs (hereafter refers to Rrs_swir), which was generated using an atmospheric correction approach for water application²³, whereas the MODIS SR was created using a land-based atmospheric correction algorithm¹. Although Rrs_swir is preferable, such a product is not readily available for global oceans. Fortunately, the MODIS SR also exhibited a good agreement with Rrs_swir in both spatial and temporal patterns in turbid inland or coastal waters²⁴. More importantly, the (see below) to invert SSC from MODIS SR product has global coverage and can be freely accessed through GEE, making

it possible for general users to perform the global application. In fact, the products.

$$SSC = \exp(0.859 * (0.145 * (R_{555}/R_{469}) + (5.167 * (R_{645}/R_{555}) * (R_{645}/(R_{645} + R_{859})) + (7.244 * (R_{859}/R_{555}) * (R_{859}/(R_{645} + R_{859}))))))^{0.990}$$

Where R_{469} , R_{555} , R_{645} , and R_{859} represent the blue, green, red, and NIR reflectance from MODIS SR has been widely used in inland and coastal waters quality monitoring 25–27 products.

We used these match-ups to conduct a recalibration for Ref. 22's algorithm. We replaced the single band reflectance (645 nm) in Ref. 22's algorithm with a band ratio ($R_{645}/555$, which is the ratio of SR between MODIS 645 and 555 nm bands) for sediment retrieval in relatively clear waters, and To further used the ratio between 859 and 645 nm ($R_{859}/645$) for sediment retrieval in highly turbid waters, creating a reliable global SSC retrieval algorithm (Extended Data Fig. 1). The algorithm was defined as follows:–

$$SSC_{Low} = 0.82 * e^{4.62 * (R_{645}/555)}$$

$$SSC_{High} = 336.23 * (R_{859}/645)^2 + 387.58 * (R_{859}/645)$$

$$SSC_{Middle} = \alpha * SSC_{Low} + \beta * SSC_{High}$$

$$SSC(\text{mg/L}) = \begin{cases} SSC_{Low} & (SSC_{Low} < 50 \text{ mg/L}) \\ SSC_{High} & (150 \text{ mg/L} < SSC_{Low}) \\ SSC_{Middle} & (50 \text{ mg/L} \leq SSC_{Low} \leq 150 \text{ mg/L}) \end{cases}$$

Similar to Feng et al. (2014), we defined the α as $\ln(150/SSC_{Low})/\ln(150/50)$, and β as $\ln(SSC_{Low}/50)/\ln(150/50)$. In this adapted algorithm, the SSC_{Low} was first calculated using $R_{645}/555$, and then the retrieval algorithm was determined based on SSC_{Low} . For the relatively clear water ($SSC < 50 \text{ mg/L}$), SSC_{Low} was adopted, and for water with $SSC > 150 \text{ mg/L}$, the SSC_{High} was selected; while for the intermediated SSC, a mixture between SSC_{Low} and SSC_{High} was applied to remove the discontinuity between these two algorithms. Extended Data Fig. 9 shows the spatial distribution of SSC derived by this algorithm in different deltas, which clearly demonstrates the smooth transition between low and high SSC. The band ratio-based algorithm used in this study has been proved to have several advantages, such as limiting the influences of the Fresnel reflection effect caused by the unique skylight reflection at the air-water interface 28, and resolving the problems resulting from variations of sediment types (grain size and density) and illumination conditions 27,29,30.–

Both the SSC_{High} and SSC_{Low} algorithms show the highest determination coefficient ($R^2 > 0.94$, $p < 0.05$), and the whole algorithm exhibited a low root mean square error (RMSE) of 28.3% (Extended Data Fig. 1). We also validate the accuracy of Yu_adapted, we first compared the minimum and maximum ranges of our MODIS-derived monthly mean SSC to the previously reported SSC ranges through a review of both published journal papers and literature. In total, 2431 deltas across the globe were examined (Supplemental Supplementary Table 12). We found observed that our MODIS-derived SSC agrees-aligns well with the published SSC for most deltas, yet discrepancies also exist in some deltas, but considering. . Considering the disagreements of the investigation time period and satellite sensor selections between our study and other publications, these discrepancies disparities are considered acceptable.

Then we applied this algorithm on– Additionally, we compared the multi-year average SSC between 500-m MODIS and 4-km Sentinel-3 OLCI, noting that the average SSC from MODIS compared to OLCI has a mean ratio of 1.41 ± 0.83 (Supplementary Figure 1e). However, when SSC exceeds 100 mg/L, the average SSC from MODIS is noticeably higher than that from OLCI. Given the differences in spatial resolution, we believe such discrepancies acceptable. Furthermore, we separately compared the temporal variations in monthly mean SSC from 2016 to 2020 for several

typical estuaries, ranging from clear to turbid. We observed that the trends based on MODIS and OLCI are very similar (Supplementary Figure 12). Consequently, we applied the Yu_adapted algorithm to MODIS Aqua and Terra 8-day SR products to retrieve coastal SSC for the 349 deltas. ~~Note that, the~~ The MODIS Terra has been noted for its radiometric degradation and calibration errors, ~~which were often considered~~ deemed unreliable in the ocean color community^{80,81}. However, ~~Ref. Hou, et al. 27 indicated that SSC derived from Terra were~~ we observed comparable ~~to that generated~~ magnitude and dynamics of SSC from Terra and Aqua ~~in a long-term changing pattern from relatively clear water (e.g., SSC < 20 mg/L)~~ across clear to turbid waters (Supplementary Figure 13), which has also been documented in previous research⁸¹ ~~water (e.g., SSC > 50 mg/L), thereby we adopted the~~. Therefore, we adopted Terra data to complement Aqua observation in this research.

2.4

Can we define one reference valid method to estimate globally SPM in coastal waters using ocean color satellite imagery? Different studies relying on different methodologies provide different results...we need to define the best reliable method.

R2.4

Thanks for raising this. In the development of ocean color remote sensing, numerous SPM algorithms have been designed for ocean color remote sensing satellites, such as the global sediment concentration monitoring algorithms proposed by Yu et al. (2019) and Wei et al. (2021). However, the applicability of these algorithms remains limited for several reasons:

(1) There is still a scarcity of field observation data that covers various water quality conditions. For example, Yu's algorithm exhibits lower accuracy in inverting SSC at medium to high levels (Yu et al. 2019), largely due to the limited number of measured samples in this SSC range;

(2) The accuracy of atmospheric correction in ocean color satellites is a crucial factor influencing algorithm precision. Although Wei's algorithm effectively inverts concentrations in clear water bodies, it typically utilizes blue light bands, which are prone to atmospheric correction errors in ocean color sensors (Wei et al. 2020), potentially introducing inaccuracies;

(3) A high spatial resolution (e.g., <1000 m) remote sensing reflectance (Rrs) product, generated using an atmospheric correction approach tailored for ocean color application (Wang and Shi 2007), is not readily available across the world, which limits monitoring capabilities for small estuaries or bays.

Therefore, in the future, as scientists worldwide continue to share measured data from diverse regions, enhance the accuracy of atmospheric correction for ocean color images, and make high-resolution ocean color Rrs products more accessible, the development of a universally applicable coastal SSC algorithm using ocean color satellite imagery may become feasible.

2.5

Your method is based on monthly composites of coastal SSC, is the monthly temporal resolution justified?

R2.5

Thanks for raising this. The reviewer is correct that some coastal dynamics occur on timescales of days or weeks, such as storms and flood. The MODIS 8-day revisit time, averaged to a monthly mean, might therefore miss some of these events. However, our statistical analysis of the monthly data shows that there are, generally, smooth transitions between months (Fig.1 in main text and Supplementary Figures 5&7), and that there are strong seasonal cycles.

We find that averaging toward a monthly basis is good, because then, most images are reliable. Given the potential for incomplete data coverage in monthly synthesis from Terra and Aqua, we assert that the monthly SSC and RPA are reliable only when the valid data in monthly images exceeds 50% (e.g., a ratio of valid pixels to the total pixels in study region). Through sensitivity analysis, we found that when the valid ratio threshold varies from 50% to 70%, less than 2% of delta SSC trends alter. However, when the threshold exceeds 80%, over 5% of delta trends change, likely due to a notable reduction in the data volume used in trend analysis. Through the synthesis of observations from Terra and Aqua, we found that, for most months between 2000 and 2020, valid observation data coverage for various estuaries exceeded 50% of the study areas (Fig. R10). Hence, the 50% threshold was adopted. Such data availability could not be achieved solely by relying on Aqua data. That is why we chose to incorporate Terra data. Nevertheless, even with the combined use of data from Terra and Aqua, certain deltas may not meet the 50% threshold in specific months (Fig. R10). In such instances, we mitigated data gaps by averaging data from preceding and subsequent months or from the same month in adjacent years. Note that for some high-latitude estuaries, there may be a lack of data during the winter and spring seasons (depicted by regularly spaced blue pixels in Fig. R10). In these estuaries, our statistical analysis is conducted solely based on the available data for certain quarters. We have included these explanations in the Methods section, please refer to Lines 803-814.

Fig. R10 | Maps of the proportion of valid data coverage (valid data coverage /estuary study area extent) for 349 estuaries over 251 months from 2000 to 2020. Each column, arranged from left to right, represents an individual estuary from west to east. Each row, from top to bottom, corresponds to each month from 2000 to 2020.

Lines 803-814:

Given the potential for incomplete data coverage in monthly synthesis from Terra and Aqua, we assert that the monthly SSC and RPA are reliable only when the valid data in monthly images exceeds 50% (e.g., a ratio of valid pixels to the total pixels in study region). Through sensitivity analysis, we found that when the valid ratio threshold varies from 50% to 70%, less than 2% of delta SSC trends alter. However, when the threshold exceeds 80%, over 5% of delta trends change, likely due to a notable reduction in the data volume used in trend analysis. We noted that, for most months between 2000 and 2020, valid coverage for the majority of deltas exceeded 50%. Hence, the 50% threshold was adopted. Nonetheless, certain deltas may not meet the 50% threshold in specific months. In such instances, we mitigated data gaps by averaging data from preceding and subsequent months or from the same month in adjacent years. It's important to mention that some high-latitude estuaries may lack data during winter and spring seasons. In such instances, our statistical analysis relies solely on available data for certain quarters.

2.6

Your first objective is not really original...the second is more ambitious

R2.6

We have modified this expression.

Lines 61-64:

We attempt to ~~address two~~ answer three fundamental questions: (i) What are the spatial and temporal patterns of ~~global~~ coastal SSC near river deltas worldwide over the past two decades? (ii) ~~What are the possible controls on global coastal SSC, and how~~ How does SSC respond to the changes in river sediment supply? (iii) What are the possible other controls on coastal SSC near river deltas?

2.7

Mapping global coastal SSC

Authors should clarify if their study focuses on SSC in river deltas or coastal zones; I understand it is focused on river deltas and does not really extend to coastal zones.

R2.7

We focus on river deltas, and we have modified the expression throughout the paper.

2.8

Can temporal trends be accurately detected combining 2 satellite sensors MODIS-T and MODIS-A? Would using only one sensor be more reliable for trend detection?

R2.8

We acknowledge that the MODIS Terra has been noted for its radiometric degradation and calibration errors, often deemed unreliable in the ocean-color community (Franz et al. 2007). Herein, we conducted a comparative analysis of the monthly average SSC obtained from MODIS Terra and Aqua using the Yu_adapted algorithm from 2002 to 2020, taking a clear estuary (Ebro) and a turbid estuary (Yangtze) as examples (Fig. R11). It shows that regardless of whether it was a clear or turbid estuary, the average ratio of Terra SSC to Aqua SSC was approximately 1.1, indicating that Terra SSC is very close to Aqua SSC. In addition, the patterns of SSC long-term dynamics in Terra and Aqua are very similar. This observation also has been documented in previous research (Hou et al. 2017). Additionally, integrating Terra data with Aqua data can increase the valid observations (Fig. R10), thereby enhancing the precision of our results. As such, we adopted Terra data to complement Aqua observation in this research. We have included these explanations in the Methods section, please refer to Lines 745-752.

Fig. R11 | Comparison of monthly mean SSC in the Yangtze and Ebro based on the Yu_adapted algorithm and MODIS Terra and Aqua SR data. The mean and standard deviation of the monthly mean ratios between Terra and Aqua, as well as the multi-year average SSC for both estuaries, are provided.

Lines 745-752:

The MODIS Terra has been noted for its radiometric degradation and calibration errors, which were often considered deemed unreliable in the ocean color community⁸⁰³⁴. However, Ref. Hou, et al. 27 indicated that SSC derived from Terra were we observed comparable to that generated magnitude and dynamics of SSC from Terra and Aqua in a long term changing pattern from relatively clear water (e.g., SSC<20 mg/L) across clear to turbid waters (Supplementary Figure 13), which has also been documented in previous research⁸¹ water (e.g., SSC>50 mg/L), thereby we adopted the. Therefore, we adopted Terra data to complement Aqua observation in this research.

2.9

Methodology: I would actually suggest the authors to review existing methods to map globally coastal SSC, identify once for all the best method, i.e., the one to use to detect changes at global and regional scales along the last two decades. This would be useful to start the study.

R2.9

Thanks for your kind suggestion again. Through the literature review, we found that currently, the most optimal algorithms for nearshore SSC inversion globally are those developed by Yu et al. (2019) and Wei et al. (2020). These two algorithms can achieve high-precision inversion of SSC through combinations of characteristic satellite spectral bands, without relying on inherent optical property parameters (e.g., backscattering coefficient), thus making it easily achievable. Both Yu and Wei have conducted comparisons between their algorithms and those of previous studies, demonstrating higher accuracy with their approaches.

In this revision, we have compared the accuracy of algorithms from Yu et al. (2019), Han et al. (2016), and Feng et al. (2014) based on our collected in-situ measured SSC and corresponding daily MODIS Aqua surface reflectance. For more details, please refer to section R2.3. Our comparison can also be found from the supplementary Note 1 in the supplementary file.

2.10

You assume your SSC retrieval algorithm is accurate according to Extended Data Fig. 1: for me this figure clearly shows your algorithm:

> is not at all accurate for $SSC < 10$ mg/L (high scatter and overestimation in Fig. 1A)

> is not calibrated with enough in situ data for $SSC > 50$ mg/L

> is not validated with enough in situ data for $SSC > 100$ mg/L

These are 3 potential serious limits in your algorithm and the main reason why you must explain why using this algorithm instead of, e.g., the one developed by Wei et al. (2021) global SPM algorithm? And how does the choice of the algorithm impact on detected trends? Or you should limit your analysis to river deltas where SSC ranges from 10 to 50 mg/L?

R2.10

We agree that our previous algorithm may have considerable uncertainty in the inversion of low SSC (Table R1). Through comparative analysis, we found that the Yu_adapted algorithm exhibits the highest accuracy, even demonstrating good precision at low concentrations ($SSC < 10$ mg/L) (Table R2). Consequently, we have adopted the Yu_adapted algorithm for SSC inversion and have updated all our SSC results accordingly. We understand that despite the use of the Yu_adapted algorithm, significant uncertainty may arise in the high SSC inversion due to limited high SSC samples. To mitigate the impact of errors on the results, we have limited our analysis to SSC below 500 mg/L only (please refer to R2.3 for details). If we were to limit it to 10-50 mg/L, many deltas would be excluded from the change monitoring due to data scarcity.

Table R2 | Accuracy of the Yu_adapted algorithm at different SSC levels. The number (n) in parentheses represents the sample size.

	Yu_adapted	
	RMSE (%)	MRE (%)
SSC<10 mg/L (n=194)	28.20	27.00
50>=SSC>=10 mg/L (n=302)	23.70	23.00
500>=SSC>=50 mg/L (n=13)	49.06	53.86
Total	24.90	24.56

2.11

MODIS surface reflectance (SR) products: despite the study by Feng et al. (2018), it is still questionable to use such a product for low turbidity deltas (e.g., the Ebro, Spain) as the atmospheric correction processing was developed for land surfaces.

R2.11

Thanks for raising this. We realized that the MODIS surface reflectance (SR) products were produced using a land-based atmospheric correction algorithm (Vermote et al. 2002), which does not correct for skylight reflection at the air-water interface (e.g., Fresnel reflection), potentially introducing errors in clear water inversion. Theoretically, the standard ocean color atmospheric correction algorithm nested within SeaDAS should be used to obtain water surface reflectance (Rrs) for retrieving SSC. However, processing global MODIS data over the past 20 years using these algorithms requires high-performance computing and storage capabilities, making global monitoring challenging.

Indeed, previous research suggests that in relatively turbid water bodies, the issue of skylight reflection can be partially alleviated through band ratio methods (Doxaran et al. 2004; Doxaran et al. 2009b). In this study, we utilized the Yu_adapted algorithm, which employs band ratios. When assessing the accuracy of SSC derived using this algorithm alongside MODIS SR against in-situ measured SSC, we observed that this algorithm achieves an accuracy of 28.2% for clear water (SSC<10 mg/L) (Table R2). This accuracy is comparable to that achieved by previous studies utilizing satellite-derived Rrs in clear water (Wei et al. 2021), which reported an accuracy of 32%. Moreover, through a comparison of SSC derived from MODIS SR with SSC obtained from OLCI in certain relatively clear estuaries (such as Ebro) (Fig. R9), we found that SSC derived from MODIS SR exhibits reliability in its trend and dynamics.

Numerous studies have retrieved SSC and monitored changes in SSC using 500-m resolution MODIS SR in both inland and coastal water (Chen et al. 2023; Hou et al. 2017; Moreno-Madrinan et al. 2010; Park and Latrubesse 2014; Wang et al. 2009). Furthermore, at present, the accessibility of MODIS SR through Google Earth Engine (GEE) enables the possibility of computing and storing data for global monitoring of estuarine SSC changes.

In the future, with further improvements in data processing and storage capabilities, we anticipate being able to estimate global SSC based on high spatial resolution Rrs derived from ocean color atmospheric correction.

2.12

Fig. 1: In addition to the classification of river deltas over the 6 continents, I would suggest to consider separately Arctic rivers as presumably more impacted by climate change effects.

R2.12

Thanks for your suggestions. In the revision, we have added analysis about the SSC, RPA, and their variations for both Arctic and non-Arctic rivers (Fig. 1 in main text and Fig. R7). Additionally, we have provided relevant descriptions in the main txt. Please refer to Lines 91-94, 118-119, 144-146, 170-173, and 308-314.

Lines 91-94:

This higher SSC can be attributed to the ~~large~~ substantial sediment loads from large rivers such as the Yangtze, Mekong, Ganges, and ~~Amazon~~ certain Arctic (with latitude $>50^{\circ}\text{N}$) deltas (e.g., Kolyma and Lena) in ~~these regions~~-this region (Fig. 1d).

Lines 118-119:

Additionally, the median RPA of Arctic deltas is over two times that of ~~SSC: 84.6 mg/L; mean~~ lower latitude deltas (Supplementary Figure 4c).

Lines 144-146:

Arctic deltas exhibited prominent increases in SSC, with a mean rate of $+1.29\text{ mg/L yr}^{-1}$, which is 18.4 times that of lower latitude deltas (Supplementary Figure 6a).

Lines 170-173:

Similar to SSC, a considerable increase in RPA was observed in Arctic deltas, with a mean rate of $+3.94\text{ km}^2\text{ yr}^{-1}$, which is far more than that in non-Arctic deltas (mean rate of $+0.23\text{ km}^2\text{ yr}^{-1}$) (Supplementary Figure 6b).

Lines 308-314:

Moreover, we found that Q_{river} , Q_{tide} , Q_{wave} , and salinity significantly ($p<0.05$) contributed to the SSC changes in 33%, 2%, 12%, and 7% of Arctic deltas, respectively (Supplementary Table 3). Despite the slight decreases in Q_{river} and Q_{tide} in Arctic deltas, the SSC of many deltas in this region has increased (Fig. 2 & Supplementary Figure 9). This might be attributed to the weaker effects from wave and salinity, which have all decreased (Supplementary Figure 9). These findings underscore the intricate responses of coastal SSCs to changes beyond river sediment supply.

2.13

Long-term trends in global coastal SSC

Global increase of +3.7% per decade, i.e., $+4.7\text{ mg/l}$ per decade: does the method used (and uncertainties associated to satellite-derived SSC) allow detecting such small variations?

R2.13

We examined how the uncertainties in satellite SSC retrievals could propagate into long-term trends as follows:

- (1) We added Gaussian-distributed random noise, with a median value of 24.9% (that is, the RMSE of the SSC retrieving algorithm, Table R2), to all time-series SSC retrievals of each delta;
- (2) We compared the long-term trends between the original and noise-added SSC.

We found that original and noise-added SSC exhibited similar change trends (either increasing or decreasing) for almost all of the deltas we examined (Fig. R12), with minor differences ($r^2=0.99$, median ratio=1.0, and median difference=0.16%) between their trends. Therefore, the uncertainty of the satellite SSC retrievals only represents random differences between satellite and in-situ measurements for individual points, rather than their systematic bias. Indeed, when many retrievals are aggregated across time and space, the uncertainty levels would be substantially reduced (Fig. R12).

Fig. R12 | Trend comparisons between original and noise-added SSC for 349 deltas.

2.14

Statistical methods exist to detect not only one but several trends over a selected time period, e.g., Muggeo (2003): estimating regression models with unknown breakpoints. *Stat. Med.* 22, 3055–3071. The authors should better explain/justify the method they adopted for the statistical analysis of deltaic coastal changes and impacts on resulting trends over two decades. For example, I can clearly see several periods in the time series generated for each continent in Extended Data Fig. 3.

R2.14

We thank the reviewer for pointing out this. In this revision, we conducted trend analysis for SSC monthly anomaly data using the Mann-Kendall test in MATLAB 2021. This method is widely used to detect trends in time series data (Da Silva et al. 2015; Hamed 2008). We have added detailed descriptions of the trend analysis, please see section R1.3.

We acknowledge the reviewer's concern regarding potential breakpoints within the time series data, which might affect trend analysis outcomes. To address this concern, we utilized a BEAST model (a Bayesian model averaging algorithm, Zhao et al. (2019)) to identify breakpoints in the continental SSC and RPA monthly anomaly time series (Figs. R13 & 14). This model can assess the relative effectiveness of individual trend decomposition models by employing Bayesian model averaging across all models. BEAST can detect change points and trends in the time series data, as well as provide credible uncertainty measures, such as the occurrence probability of the change points. The model developer has generously provided MATLAB and R packages for direct utilization.

Based on the trend analysis results, it seems that only approximately half of the continents can detect breakpoints in either SSC or RPA, and the occurrence probability of these breakpoints is not

notably high (e.g., probabilities exceeding 50%). Upon examining the trend lines of SSC and RPA for each continent, it is evident that all continents exhibit an overall increasing trend in SSC and RPA, except for Asia. These findings align with the trends obtained from the Mann-Kendall test (Supplementary Figure 5). Therefore, we opt to utilize the results from the Mann-Kendall test in the main text.

Fig. R13 | Trend breakpoint analysis of monthly anomaly in SSC across six continents. A BEAST model (a Bayesian model averaging algorithm) was used to detect breakpoints. The breakpoints detection results from BEAST are displayed into four panels (Trend, probability, tOrder, and Error). Trend panel shows the monthly anomaly data and the trend line (dark grey line); the probability panel indicates the occurrence probability of trend points; the tOrder panel indicates orders of polynomials used to model the trend; and error panel exhibited the model residuals. The red line in each panel shows the possible breakpoints. The analysis was conducted in MATLAB 2021.

Fig. R14 | Trend breakpoint analysis of monthly anomaly in RPA across six continents. A BEAST model (a Bayesian model averaging algorithm) was used to detect breakpoints. The breakpoints detection results from BEAST are displayed into four panels (Trend, probability, tOrder, and Error). Trend panel shows the monthly anomaly data and the trend line (dark grey line); the probability panel indicates the occurrence probability of trend points; the tOrder panel indicates orders of polynomials used to model the trend; and error panel exhibited the model residuals. The red line in each panel shows the possible breakpoints. The analysis was conducted in MATLAB 2021.

2.15

Fig. 2: the figure legends and/or caption should be changed as we do not expect negatives values for SSC (mg/L) no RPA (km²).

R2.15

We have modified the label of Y-axes in Fig. 2b.

2.16

Controls on coastal SSC

Fig. 3: data points could be colored to distinguish the different regions of interest, i.e., the 6 different continents.

R2.16

We have modified the figure, please see Fig. 4.

2.17

Could the authors also present/analyze the time series of Q_{river} , Q_{tide} , Q_{wave} and salinity globally and per continent?

R2.17

We have added a figure (Fig. R15) and corresponding analysis to show the changes of Q_{river} , Q_{tide} , Q_{wave} , and Salinity, please see Lines 305-308, 310-313, and Supplementary Figure 9.

Lines 305-308:

Increases in Q_{river} and Q_{tide} along the coasts of Atlantic coast of the America contributed most to SSC increases there, whereas their decreases, coupled with increases in Q_{wave} and salinity likely led to declines in SSC decrease in South Asia (Fig. 4e & Supplementary Figure 9).

Lines 310-313:

Despite the slight decreases in Q_{river} and Q_{tide} in Arctic deltas, the SSC of many deltas in this region has increased (Fig. 2 & Supplementary Figure 9). This might be attributed to the weaker effects from wave and salinity, which have all decreased (Supplementary Figure 9).

Fig. R15 | Long-term changes of Q_{river} , Q_{tide} , Q_{wave} , and salinity of global deltas. (a-c). Distribution of long-term annual changes of Q_{river} , Q_{tide} , and Q_{wave} in 186 river deltas. Trends were analyzed from 2000 to 2020; (d). Long-term variations of salinity for 139 deltas from 2000 to 2020. The change rate is represented by Sen's slope from the Mann-Kendall test. The left chart of each panel displays the number of deltas with different changing trends. Sig. Inc.: significant ($p < 0.05$) increase; Sig. Dec.: significant ($p < 0.05$) decrease; Inc.: Increase; Dec.: Decrease.

Reference

1. Balasubramanian, S.V., Pahlevan, N., Smith, B., Binding, C., Schalles, J., Loisel, H., Gurlin, D., Greb, S., Alikas, K., & Randla, M. (2020). Robust algorithm for estimating total suspended solids (TSS) in inland and nearshore coastal waters. *Remote Sensing of Environment*, 246, 111768
2. Bates, C.C. (1953). Rational theory of delta formation. *Aapg Bulletin*, 37, 2119-2162
3. Becherer, J., Flüser, G., Umlauf, L., & Burchard, H. (2016). Estuarine circulation versus tidal pumping: Sediment transport in a well-mixed tidal inlet. *Journal of Geophysical Research: Oceans*, 121, 6251-6270
4. Bednarek, A.T. (2001). Undamming rivers: a review of the ecological impacts of dam removal. *Environmental management*, 27, 803-814
5. Brenon, I., & Le Hir, P. (1999). Modelling the turbidity maximum in the Seine estuary (France): identification of formation processes. *Estuarine, Coastal and Shelf Science*, 49, 525-544
6. Cai, X., Feng, L., Hou, X., & Chen, X. (2016). Remote Sensing of the Water Storage Dynamics of Large Lakes and Reservoirs in the Yangtze River Basin from 2000 to 2014. *Scientific Reports*, 6, 1-9
7. Cao, B., Qiu, J., Zhang, W., Xie, X., Lu, X., Yang, X., & Li, H. (2022). Retrieval of suspended sediment concentrations in the Pearl River estuary using multi-source satellite imagery. *Remote Sensing*, 14, 3896
8. Chen, R., Jiang, X., & Chen, J. (2023). Suspended sediment change along the Zhejiang-Fujian coast revealed using MODIS 250-m imagery. *Estuarine, Coastal and Shelf Science*, 293, 108511
9. Da Silva, R.M., Santos, C.A., Moreira, M., Corte-Real, J., Silva, V.C., & Medeiros, I.C. (2015). Rainfall and river flow trends using Mann–Kendall and Sen’s slope estimator statistical tests in the Cobres River basin. *Natural Hazards*, 77, 1205-1221
10. Dethier, E.N., Renshaw, C.E., & Magilligan, F.J. (2022). Rapid changes to global river suspended sediment flux by humans. *Science*, 376, 1447-1452
11. Dogliotti, A.I., Ruddick, K., Nechad, B., Doxaran, D., & Knaeps, E. (2015). A single algorithm to retrieve turbidity from remotely-sensed data in all coastal and estuarine waters. *Remote Sensing of Environment*, 156, 157-168
12. Doxaran, D., Cherukuru, R.N., & Lavender, S. (2004). Estimation of surface reflection effects on upwelling radiance field measurements in turbid waters. *Journal of Optics A: Pure and Applied Optics*, 6, 690
13. Doxaran, D., Froidefond, J.-M., Castaing, P., & Babin, M. (2009a). Dynamics of the turbidity maximum zone in a macrotidal estuary (the Gironde, France): Observations from field and MODIS satellite data. *Estuarine, Coastal and Shelf Science*, 81, 321-332
14. Doxaran, D., Froidefond, J.-M., Castaing, P., & Babin, M. (2009b). Dynamics of the turbidity maximum zone in a macrotidal estuary (the Gironde, France): Observations from field and MODIS satellite data. *Estuarine, Coastal and Shelf Science*, 81, 321-332
15. Dyer, K.R. (1973). *Estuaries: a physical introduction*.
16. Edmonds, D.A., Toby, S.C., Siverd, C.G., Twilley, R., Bentley, S.J., Hagen, S., & Xu, K. (2023). Land loss due to human-altered sediment budget in the Mississippi River Delta. *Nature Sustainability*, 1-8

17. Eisma, D., & Kalf, J. (1984). Dispersal of Zaire River suspended matter in the estuary and the Angola Basin. *Netherlands Journal of Sea Research*, *17*, 385-411
18. Feng, L., Hu, C., Chen, X., & Song, Q. (2014). Influence of the Three Gorges Dam on total suspended matters in the Yangtze Estuary and its adjacent coastal waters: Observations from MODIS. *Remote Sensing of Environment*, *140*, 779-788
19. Flores, R.P., Williams, M.E., & Horner-Devine, A.R. (2022). River plume modulation by infragravity wave forcing. *Geophysical Research Letters*, *49*, e2021GL097467
20. Franz, B.A., Kwiatkowska, E.J., Meister, G., & McClain, C.R. (2007). Utility of MODIS-Terra for ocean color applications. *Earth Observing Systems XII*, 6677, 279-292
21. Garc ía Berdeal, I., Hickey, B., & Kawase, M. (2002). Influence of wind stress and ambient flow on a high discharge river plume. *Journal of Geophysical Research: Oceans*, *107*, 13-11-13-24
22. Geyer, W.R., & MacCready, P. (2014). The estuarine circulation. *Annual Review of Fluid Mechanics*, *46*, 175-197
23. Hamed, K.H. (2008). Trend detection in hydrologic data: the Mann–Kendall trend test under the scaling hypothesis. *Journal of hydrology*, *349*, 350-363
24. Han, B., Loisel, H., Vantrepotte, V., M ériaux, X., Bry ère, P., Ouillon, S., Dessailly, D., Xing, Q., & Zhu, J. (2016). Development of a Semi-Analytical Algorithm for the Retrieval of Suspended Particulate Matter from Remote Sensing over Clear to Very Turbid Waters. *Remote Sensing*, *8*(211)
25. He, Z., Xu, B., Okon, S.U., & Li, L. (2022). Numerical Investigation of the Sediment Hyperpycnal Flow in the Yellow River Estuary. *Journal of Marine Science and Engineering*, *10*, 943
26. Horner-Devine, A.R., Hetland, R.D., & MacDonald, D.G. (2015). Mixing and transport in coastal river plumes. *Annual Review of Fluid Mechanics*, *47*, 569-594
27. Hou, X., Feng, L., Duan, H., Chen, X., Sun, D., & Shi, K. (2017). Fifteen-year monitoring of the turbidity dynamics in large lakes and reservoirs in the middle and lower basin of the Yangtze River, China. *Remote Sensing of Environment*, *190*, 107-121
28. Kemp, D.B., Sadler, P.M., & Vanacker, V. (2020). The human impact on North American erosion, sediment transfer, and storage in a geologic context. *Nature Communications*, *11*, 1-9
29. Kirwan, M.L., Temmerman, S., Skeeahan, E.E., Guntenspergen, G.R., & Fagherazzi, S. (2016). Overestimation of marsh vulnerability to sea level rise. *Nature Climate Change*, *6*, 253-260
30. Liang, Y.-C., Lo, M.-H., Lan, C.-W., Seo, H., Ummenhofer, C.C., Yeager, S., Wu, R.-J., & Steffen, J.D. (2020). Amplified seasonal cycle in hydroclimate over the Amazon river basin and its plume region. *Nature Communications*, *11*, 4390
31. Loisel, H., Mangin, A., Vantrepotte, V., Dessailly, D., Dinh, D.N., Garnesson, P., Ouillon, S., Lefebvre, J.-P., M ériaux, X., & Phan, T.M. (2014). Variability of suspended particulate matter concentration in coastal waters under the Mekong's influence from ocean color (MERIS) remote sensing over the last decade. *Remote Sensing of Environment*, *150*, 218-230
32. Lovelock, C.E., Cahoon, D.R., Friess, D.A., Guntenspergen, G.R., Krauss, K.W., Reef, R., Rogers, K., Saunders, M.L., Sidik, F., & Swales, A. (2015). The vulnerability of Indo-Pacific mangrove forests to sea-level rise. *Nature*, *526*, 559-563
33. Luo, W., Shen, F., He, Q., Cao, F., Zhao, H., & Li, M. (2022). Changes in suspended sediments in the Yangtze River Estuary from 1984 to 2020: Responses to basin and estuarine engineering

- constructions. *Science of the Total Environment*, 805, 150381
34. Mitchell, S. (2013). Turbidity maxima in four macrotidal estuaries. *Ocean & Coastal Management*, 79, 62-69
 35. Moreno-Madrinan, M.J., Al-Hamdan, M.Z., Rickman, D.L., & Muller-Karger, F.E. (2010). Using the surface reflectance MODIS Terra product to estimate turbidity in Tampa Bay, Florida. *Remote Sensing*, 2, 2713-2728
 36. Park, E., & Latrubesse, E.M. (2014). Modeling suspended sediment distribution patterns of the Amazon River using MODIS data. *Remote Sensing of Environment*, 147, 232-242
 37. Pritchard, M., & Huntley, D.A. (2006). A simplified energy and mixing budget for a small river plume discharge. *Journal of Geophysical Research: Oceans*, 111
 38. Russell, P.E. (1993). Mechanisms for beach erosion during storms. *Continental Shelf Research*, 13, 1243-1265
 39. Schuerch, M., Spencer, T., Temmerman, S., Kirwan, M.L., Wolff, C., Lincke, D., McOwen, C.J., Pickering, M.D., Reef, R., & Vafeidis, A.T. (2018). Future response of global coastal wetlands to sea-level rise. *Nature*, 561, 231-234
 40. Shanmugam, G. (2018). The hyperpycnite problem. *Journal of Palaeogeography*, 7, 1-42
 41. Shi, W., & Wang, M. (2010). Satellite observations of the seasonal sediment plume in central East China Sea. *Journal of Marine systems*, 82, 280-285
 42. Sommerfield, C.K., & Wong, K.C. (2011). Mechanisms of sediment flux and turbidity maintenance in the Delaware Estuary. *Journal of Geophysical Research: Oceans*, 116
 43. Syvitski, J.P., Vörösmarty, C.J., Kettner, A.J., & Green, P. (2005). Impact of humans on the flux of terrestrial sediment to the global coastal ocean. *Science*, 308, 376-380
 44. Uncles, R., & Stephens, J. (1989). Distributions of suspended sediment at high water in a macrotidal estuary. *Journal of Geophysical Research: Oceans*, 94, 14395-14405
 45. Van Maren, D., Oost, A., Wang, Z.B., & Vos, P. (2016). The effect of land reclamations and sediment extraction on the suspended sediment concentration in the Ems Estuary. *Marine Geology*, 376, 147-157
 46. Van Maren, D.S., Maushake, C., Mol, J.-W., Van Keulen, D., Jürges, J., Vroom, J., Schuttelaars, H., Gerkema, T., Schulz, K., & Badewien, T.H. (2023). Synoptic observations of sediment transport and exchange mechanisms in the turbid Ems Estuary: the EDoM campaign. *Earth System Science Data*, 15, 53-73
 47. Vermote, E.F., El Saleous, N.Z., & Justice, C.O. (2002). Atmospheric correction of MODIS data in the visible to middle infrared: first results. *Remote Sensing of Environment*, 83, 97-111
 48. Wang, F., Zhou, B., Xu, J., Song, L., & Wang, X. (2009). Application of neural network and MODIS 250m imagery for estimating suspended sediments concentration in Hangzhou Bay, China. *Environmental Geology*, 56, 1093-1101
 49. Wang, M., & Shi, W. (2007). The NIR-SWIR combined atmospheric correction approach for MODIS ocean color data processing. *Optics Express*, 15(24), 15722-15733
 50. Wei, J., Wang, M., Jiang, L., Yu, X., Mikelsons, K., & Shen, F. (2021). Global estimation of suspended particulate matter from satellite ocean color imagery. *Journal of Geophysical Research: Oceans*, 126, e2021JC017303
 51. Wei, J., Yu, X., Lee, Z., Wang, M., & Jiang, L. (2020). Improving low-quality satellite remote sensing reflectance at blue bands over coastal and inland waters. *Remote Sensing of Environment*, 250, 112029

52. Xie, D., Schwarz, C., Brückner, M.Z., Kleinhans, M.G., Urrego, D.H., Zhou, Z., & Van Maanen, B. (2020). Mangrove diversity loss under sea-level rise triggered by bio-morphodynamic feedbacks and anthropogenic pressures. *Environmental Research Letters*, 15, 114033
53. Yu, Q., Wang, Y., Gao, J., Gao, S., & Flemming, B. (2014). Turbidity maximum formation in a well-mixed macrotidal estuary: The role of tidal pumping. *Journal of Geophysical Research: Oceans*, 119, 7705-7724
54. Yu, X., Lee, Z., Shen, F., Wang, M., Wei, J., Jiang, L., & Shang, Z. (2019). An empirical algorithm to seamlessly retrieve the concentration of suspended particulate matter from water color across ocean to turbid river mouths. *Remote Sensing of Environment*, 235, 111491
55. Zhao, K., Wulder, M.A., Hu, T., Bright, R., Wu, Q., Qin, H., Li, Y., Toman, E., Mallick, B., & Zhang, X. (2019). Detecting change-point, trend, and seasonality in satellite time series data to track abrupt changes and nonlinear dynamics: A Bayesian ensemble algorithm. *Remote Sensing of Environment*, 232, 111181

REVIEWERS' COMMENTS

Reviewer #1 (Remarks to the Author):

The authors have done a great deal of detailed, well-documented work to improve their article. The main points requiring clarification have been incorporated.

What remains is the basic assumption that sediment fluxes are estimated by multiplying water discharge by surface concentration. Mixing in macrotidal estuaries allows for this, but the assumption deserves validation and discussion in micro- or meso-tidal estuaries, where stratification and 3D effects are more important. A comment can be added on this subject in the final version of the article: macrotidal estuaries lend themselves better to flow analysis by remote sensing than meso- or microtidal estuaries, where 3D effects can considerably modulate flows. The combined use of remote sensing and modelling could offer an alternative to the analysis of these systems.

The replacement of unpublished data from Sun et al. with those from Dethier et al (2022) fully addresses an important weakness in the original version. The authors have dug deeper and made some very instructive comparisons in the response to reviewer R1.5. The difference between MODIS SSC and Landsat SSC (Fig R6 a) is still very significant, on the order of an order of magnitude, at high concentrations: Landsat SSC is between 800 and 1000 mg/L, while MODIS SSC varies between around 20 and 200 mg/L. I understand that the authors don't have the space to detail everything in their article, but I suggest adding a general comment pointing out the significant difference in SSC estimates between the two studies, which cannot be attributed to the difference in river size/flow rate alone.

With these two reservations, I give my approval to the publication of the article and congratulate the authors on their considerable work.

Sylvain OUILLON

Reviewer #2 (Remarks to the Author):

Interesting and useful study, which represents an impressive amount of work.

The authors still do not fully explain how the processes and mechanisms involved did control the changes observed in suspended sediment concentrations near river deltas. However, significant efforts were made to clarify several methodological issues and improve the analysis of results obtained. Therefore I recommend publication of this manuscript after minor revision (see my detailed comments hereafter).

Answers to Reviewer#1

Major flaw = processes and mechanisms involved

SSC and RPA have only secondary relationships and no reason to co-evolve, especially between sites.

> Qriver, Qtide, Qwave, and salinity are considered as driving factors of RPA and SSC. What about wind (strength and direction) which has a direct impact on RPA? Authors simply admit they did not explore the impact of wind as driving factor, which is a shame and confirms the limit of the study highlighted by Reviewer#1.

> Fig. R4: I think what you call Parana River is actually the Rio de la Plata.

Error in figure 2b: wrong units

> Authors have clarified this issue

Separate analysis of trends by delta morphology type

> Authors convincingly accounted for this comment (Fig. 1 and Fig. R7)

Answers to Reviewer#2

Would your results be similar or different if using different SPM algorithm?

> The effort made by authors to estimate the impact of the SSC algorithm used on RPA and SSC trends is appreciated. Notably efforts made to compare 'their' SSC to in-situ measured and OLCI satellite SSC products. Overall, the answer provides confidence in the trends presented in the study, while not on quantitative values obtained (Figs. R8 and R9).

2.8 Can temporal trends be accurately detected combining 2 satellite sensors MODIS-T and MODISA? Would using only one sensor be more reliable for trend detection?

> Thank you for providing Figure R11 which convinced me MODIS-Aqua and Aqua observations can be combined

Consider separately Arctic rivers as presumably more impacted by climate change effects

> Thank you, it clearly highlights a different behavior of Arctic rivers. Fig. 1 in main text: what is the color code (blue, orange and green circles)? Same question for Fig. 3.

Long-term trends in global SSC detected despite uncertainties associated to satellite-derived SSC

> Thank you for providing trend comparisons between original and noise-added SSC (Fig. R12)

Trend breakpoint analysis

> Figs. R13 and R14 are definitely useful to highlight that most continents did experiment several periods over the last two decades.

Reviewer #2 (Remarks on code availability):

It is partly out of my expertise to review the codes written in different programming languages by the authors. Reviewing codes on top of the manuscript would have been really time confusing, so that I preferred to focus on the manuscript. I hope other reviewer(s) could review the code.

Color Key

Black: Reviewers' comments

Blue: Author's response and the revised text from the previous manuscript. Note that the line numbers mentioned below refer to the lines in the revised Word document (e.g., Lines 335-340).

REVIEWER COMMENTS

Reviewer #1 (Remarks to the Author):

The authors have done a great deal of detailed, well-documented work to improve their article. The main points requiring clarification have been incorporated.

We thank Dr. Ouillon for the positive words.

What remains is the basic assumption that sediment fluxes are estimated by multiplying water discharge by surface concentration. Mixing in macrotidal estuaries allows for this, but the assumption deserves validation and discussion in micro- or meso-tidal estuaries, where stratification and 3D effects are more important. A comment can be added on this subject in the final version of the article: macrotidal estuaries lend themselves better to flow analysis by remote sensing than meso- or microtidal estuaries, where 3D effects can considerably modulate flows. The combined use of remote sensing and modelling could offer an alternative to the analysis of these systems.

Thanks for your suggestion. We have incorporated this clarification, please refer to Lines 336-341.

Lines 336-341:

Note that the Q_{river} employed in this study, derived from remote sensing surface SSC and water discharge, may be more applicable to the well-mixed macro-tidal estuaries. This is because, in meso- or micro-tidal estuaries, three-dimensional effects may significantly modulate river flow, introducing uncertainties in Q_{river} analysis. In the future, to obtain a more accurate Q_{river} in these systems, the combination of remote sensing and modelling could be considered as an alternative approach.

The replacement of unpublished data from Sun et al. with those from Dethier et al (2022) fully addresses an important weakness in the original version. The authors have dug deeper and made some very instructive comparisons in the response to reviewer R1.5. The difference between MODIS SSC and Landsat SSC (Fig R6 a) is still very significant, on the order of an order of magnitude, at high concentrations: Landsat SSC is between 800 and 1000 mg/L, while MODIS SSC varies between around 20 and 200 mg/L. I understand that the authors don't have the space to detail everything in their article, but I suggest adding a general comment pointing out the significant difference in SSC estimates between the two studies, which cannot be attributed to the difference in river size/flow rate alone.

Thanks for your suggestion. We have added more clarifications, please refer to Lines 331-335.

Lines 331-335:

The Landsat-derived SSC mainly originates from pixels in the river channels, which may be

several times larger than our coastal SSC. Such a significant discrepancy arises partly due to the size and flow rate of the river, and also because of the different areas estimated (e.g., river channels vs river coasts), which have different driving forces.

With these two reservations, I give my approval to the publication of the article and congratulate the authors on their considerable work.

Sylvain OUILLON

We really appreciate the valuable suggestions and recognition from Dr. Ouillon.

Reviewer #2 (Remarks to the Author):

Interesting and useful study, which represents an impressive amount of work.

The authors still do not fully explain how the processes and mechanisms involved did control the changes observed in suspended sediment concentrations near river deltas. However, significant efforts were made to clarify several methodological issues and improve the analysis of results obtained. Therefore I recommend publication of this manuscript after minor revision (see my detailed comments hereafter).

We appreciate the reviewer's positive feedback on our efforts.

Comments on Answers to Reviewer#1

Major flaw = processes and mechanisms involved

SSC and RPA have only secondary relationships and no reason to co-evolve, especially between sites.

- **Q_{river} , Q_{tide} , Q_{wave} , and salinity are considered as driving factors of RPA and SSC. What about wind (strength and direction) which has a direct impact on RPA? Authors simply admit they did not explore the impact of wind as driving factor, which is a shame and confirms the limit of the study highlighted by Reviewer#1.**

Thanks for raising this. We have further clarified the influences of wind on RPA and SSC, and have modified the expressions in the previous version as “Other factors, such as the strength and direction of the wind (García Berdeal et al. 2002; Li et al. 2010; Wang et al. 2018), could also modulate the horizontal extent of river plumes and obscure the relation between SSC magnitude and RPA. For instance, strong winds could result in significantly high SSC and an extensive RPA near the Yangtze estuary. In contrast, weak winds might lead to high SSC near the shore but a smaller RPA outside the river mouth, primarily due to the limited spread of turbid water (Li et al. 2010). Additionally, winds blowing towards the shore (e.g., eastern wind) can lead to extremely high SSC near the coast and a constrained RPA in the Yangtze estuary, while offshore winds (e.g., southwestern wind) may result in relatively low SSC yet a broader RPA (Wang et al. 2018)”. Please refer to Lines 219-227.

Lines 219-227:

Other factors, such as the strength and direction of the wind ⁴⁶⁻⁴⁸, could also modulate the horizontal extent of river plumes and obscure the relation between SSC magnitude and RPA. For instance, strong winds could result in significantly high SSC and an extensive RPA near the Yangtze estuary. In contrast, weak winds might lead to high SSC near the shore but a smaller RPA outside the river mouth, primarily due to the limited spread of turbid water ⁴⁷. Additionally, winds blowing towards the shore (e.g., eastern wind) can lead to extremely high SSC near the coast and a constrained RPA in the Yangtze estuary, while offshore winds (e.g., southwestern wind) may result in relatively low SSC yet a broader RPA ⁴⁸.

- ***Fig. R4: I think what you call Parana River is actually the Rio de la Plata.***

Thanks for your suggestion. We have made modifications accordingly throughout the paper.

Error in figure 2b: wrong units

- ***Authors have clarified this issue***

Thank you for recognizing our efforts.

Separate analysis of trends by delta morphology type

- ***Authors convincingly accounted for this comment (Fig. 1 and Fig. R7)***

Thank you for recognizing our efforts.

Answers to Reviewer#2

Would your results be similar or different if using different SPM algorithm?

- **The effort made by authors to estimate the impact of the SSC algorithm used on RPA and SSC trends is appreciated. Notably efforts made to compare ‘their’ SSC to in-situ measured and OLCI satellite SSC products. Overall, the answer provides confidence in the trends presented in the study, while not on quantitative values obtained (Figs. R8 and R9).**

Thanks for your encouraging words.

2.8 Can temporal trends be accurately detected combining 2 satellite sensors MODIS-T and MODISA?

Would using only one sensor be more reliable for trend detection?

- **Thank you for providing Figure R11 which convinced me MODIS-Aqua and Aqua observations can be combined**

Thanks for your recognition of our efforts.

Consider separately Arctic rivers as presumably more impacted by climate change effects

- **Thank you, it clearly highlights a different behavior of Arctic rivers. Fig. 1 in main text: what is the color code (blue, orange and green circles)? Same question for Fig. 3.**

Thanks for raising this. The color in Fig. 1 indicates different levels of SSC, while different colors in Fig. 3 represent different combinations of Q_{river} and SSC trends (Inc.: increase; Dec.: decrease). We have clarified these expressions in the captions of Figs. 1 and 3. Please refer to Lines 764-766, 784-785.

Lines 764-766:

The different circle sizes represent varying magnitudes of water discharge, while the different circle colors indicate different levels of SSC;

Lines 784-785:

The different colors represent different combinations of Q_{river} and SSC trends (Inc.: increase; Dec.: decrease);

Long-term trends in global SSC detected despite uncertainties associated to satellite-derived SSC

- **Thank you for providing trend comparisons between original and noise-added SSC (Fig. R12)**

Thanks for your recognition of our efforts.

Trend breakpoint analysis

- **Figs. R13 and R14 are definitely useful to highlight that most continents did experiment several periods over the last two decades.**

Thanks for your positive words.

References:

1. García Berdeal, I., Hickey, B., & Kawase, M. (2002). Influence of wind stress and ambient flow on a high discharge river plume. *Journal of Geophysical Research: Oceans*, 107, 13-11-13-24
2. Li, J., Gao, S., & Wang, Y. (2010). Delineating suspended sediment concentration patterns in surface waters of the Changjiang Estuary by remote sensing analysis. *Acta oceanologica sinica*, 29, 38-47
3. Wang, L., Zhou, Y., & Shen, F. (2018). Suspended sediment diffusion mechanisms in the Yangtze Estuary influenced by wind fields. *Estuarine, Coastal and Shelf Science*, 200, 428-436